# ANNOTATION-EFFICIENT LANGUAGE MODEL ALIGNMENT VIA DIVERSE AND REPRESENTATIVE RESPONSE TEXTS

## ABSTRACT

Preference optimization is a standard approach to fine-tuning large language models to align with human preferences. The quantity, diversity, and representativeness of the preference dataset are critical to the effectiveness of preference optimization. However, obtaining a large amount of preference annotations is difficult in many applications. This raises the question of how to use the limited annotation budget to create an effective preference dataset. To this end, we propose Annotation-Efficient Preference Optimization (AEPO). Instead of exhaustively annotating preference over all available response texts, AEPO selects a subset of responses that maximizes diversity and representativeness from the available responses and then annotates preference over the selected ones. In this way, AEPO focuses the annotation budget on labeling preferences over a smaller but informative subset of responses. We evaluate the performance of Direct Preference Optimization (DPO) using AEPO and show that it outperforms models trained using a standard DPO with the same annotation budget. Our code is available at https://anonymous.4open.science/r/aepo-05B2.

## 1 INTRODUCTION

Large Language Models (LLMs) trained on massive datasets are capable of solving a variety of tasks in natural language understanding and generation (Brown et al., 2020; Ouyang et al., 2022; Touvron et al., 2023; OpenAI et al., 2024). However, they have been shown to generate texts containing toxic, untruthful, biased, and harmful outputs (Bai et al., 2022; Lin et al., 2022; Touvron et al., 2023; Casper et al., 2023; Huang et al., 2024b; Guan et al., 2024). Language model alignment aims to address these issues by guiding LLMs to generate responses that aligns with human preferences, steering them to generate responses that are informative, harmless, and helpful (Christiano et al., 2017; Ziegler et al., 2020; Stiennon et al., 2020; Bai et al., 2022).

The common strategies to align an LLM are Reinforcement learning from human feedback (RLHF) and Direct Preference Optimization (DPO) (Stiennon et al., 2020; Ouyang et al., 2022; Rafailov et al., 2023). RLHF and DPO use the human preference dataset to train a reward model or a language model directly. The performance of these algorithms is highly dependent on the choice of the preference dataset. However, building a human preference dataset requires human annotations, which are expensive to collect. Thus, the main bottleneck in building a preference dataset is the annotation cost.

A large number of works have investigated the synthesis of preference data using a powerful LLM (e.g., GPT-4) to distill the knowledge of human preferences (Dubois et al., 2023; Lee et al., 2024; Ding et al., 2023; Honovich et al., 2023; Cui et al., 2023; Mukherjee et al., 2023; Xu et al., 2024a; Liu et al., 2024a). However, human preferences are known to be diverse and pluralistic, and they are unlikely to be represented by the opinion of a single model (Qiu et al., 2022; Kirk et al., 2023; Wan et al., 2023; Cao et al., 2023b; Zhou et al., 2024; Sorensen et al., 2024a; Rao et al., 2024; Xu et al., 2024b; Sorensen et al., 2024b; Kirk et al., 2024; Shen et al., 2024b; Chakraborty et al., 2024). Several papers have pointed out that LLMs may exhibit bias toward aligning with people from a particular background (Santurkar et al., 2023; Naous et al., 2024; Adilazuarda et al., 2024). For example, Cao et al. (2023b) reports that ChatGPT has a strong alignment with American culture,

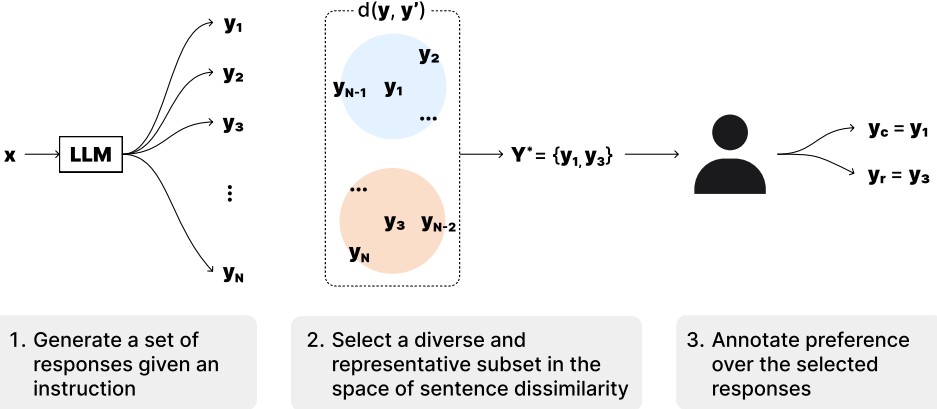

Figure 1: Annotation-Efficient Preference Optimization (AEPO) is a process for generating a preference dataset with diverse and representative responses with fewer annotations. See Section 3 for details. Here we set $k = 2$ and select two responses from the generated responses to annotate.

but adapts less effectively to other cultural contexts. In addition to cultural biases, previous work suggests that even a highly capable model (e.g., GPT-4) still has biases such as length bias (Jain et al., 2024; Dubois et al., 2024), style bias (Gudibande et al., 2024), and positional bias (Zheng et al., 2023). Thus, human annotation is desirable to align and personalize an LLM with diverse and unbiased human preferences (Greene et al., 2023; Jang et al., 2023; Kirk et al., 2023). The efficiency of annotation is critical to making LLMs accessible and useful to people from diverse backgrounds, who may have only a small amount of preference feedback data to work with.

The question is how to generate an effective preference dataset with a limited annotation budget. Previous work has shown that the following three features are desirable for a preference dataset to be effective (Liu et al., 2024c;a):

1. *Quantity and Diversity of instructions.* Greater quantity and diversity are desirable for the instruction set (Askell et al., 2021; Wang et al., 2023; Ding et al., 2023; Honovich et al., 2023; Cao et al., 2023a; Yuan et al., 2023; Yu et al., 2023; Xu et al., 2024a; Zhang et al., 2024a; Ge et al., 2024).

2. *Diversity of responses.* A set of responses with higher diversity is desirable (Cui et al., 2023; Lu et al., 2024; Yuan et al., 2023; Song et al., 2024).

3. *Representativeness of responses.* Responses that represent the behavior of the training model are more desirable (Guo et al., 2024; Tajwar et al., 2024; Tang et al., 2024a)

To achieve all three desiderata with a limited annotation budget, it is desirable to annotate preference over diverse and representative responses with a minimum amount of annotation required per instruction.

To this end, we propose **Annotation-Efficient Preference Optimization (AEPO)**, a preference optimization with a preprocessing step on the preference dataset to reduce the required amount of annotation (Figure 1). Instead of annotating the preference over all $N$ responses, AEPO selects $k(< N)$ responses from $N$ responses. We deploy a sophisticated method to select a set of response texts with high diversity and representativeness. It then annotates the preference for the selected $k$ responses. In this way, AEPO uses all $N$ samples to select a subset of responses with high diversity and representativeness, while requiring only an annotation over a subset of responses.

The strength of AEPO is threefold (Table 1). First, it is applicable to human feedback data. Compared to Reinforcement Learning from AI Feedback (RLAIF) (Lee et al., 2024), our approach can be applied to both human and AI feedback. RLAIF is a scalable approach in terms of both instructions and annotations, but it is known that the feedback from existing language models is biased in various ways (Cao et al., 2023b; Zheng et al., 2023; Jain et al., 2024; Gudibande et al., 2024; Dubois et al., 2024). Second, it is scalable with additional computational resources. By generating a larger amount of responses, AEPO can find more diverse and representative responses to annotate, result-

Table 1: Comparison of annotation strategies for preference dataset.

| Preference dataset | Human feedback | Scalable | Annotation-efficient |
|---|:---:|:---:|:---:|
| Human feedback | ✓ | ✗ | ✗ |
| RLAIF (Lee et al., 2024) | ✗ | ✓ | ✓ |
| West-of-N (Pace et al., 2024) | ✓ | ✓ | ✗ |
| **AEPO (Proposed)** | ✓ | ✓ | ✓ |

ing in a more effective preference dataset with a fixed amount of annotation (Figure 3). Third, less annotation is required to generate an effective preference dataset. Unlike an exhaustive annotation strategy which requires a large annotation effort (e.g., West-of-N strategy, Xu et al. 2023; Yuan et al. 2024b; Pace et al. 2024), AEPO can reduce the annotation cost through the subsampling process.

We evaluate the performance of DPO using AEPO on the AlpacaFarm, Anthropic's hh-rlhf, and JCommonsensMorality datasets in Section 4 (Bai et al., 2022; Dubois et al., 2023; Takeshita et al., 2023). With a fixed annotation budget, the performance of vanilla DPO degrades as the number of responses per instruction increases above a certain threshold (Figure 3). In contrast, AEPO scales with the number of responses under a fixed annotation budget, outperforming vanilla DPO when a large number of responses are available. We conduct ablation studies and observe that AEPO consistently outperforms WoN with varying settings and hyperparameters (Appendix D). The result shows that AEPO is a promising algorithm for efficient preference optimization, especially when annotation cost is the bottleneck of the alignment process.

## 2 BACKGROUND

**Preference Optimization.** Let $\mathcal{D}_p$ be a pairwise preference dataset $\mathcal{D}_p = \{(x, y_c, y_r)\}$, where $x$ is an instruction ($x \in \mathcal{X}$), $y_c$ is the chosen response, and $y_r$ is the rejected response, that is, $y_c$ is preferred to $y_r$ ($y_c, y_r \in \mathcal{Y}$). One of the popular algorithms for learning from the preference dataset is **Direct Preference Optimization (DPO)** (Rafailov et al., 2023). DPO trains the language model to directly align with the human preference data over the responses without using reward models. The objective function of the DPO is the following:

$$\pi_{\text{DPO}} = \arg\max_{\pi} \mathbb{E}_{(x,y_c,y_r)\sim\mathcal{D}_p} [\log \sigma(\beta \log \frac{\pi(y_c|x)}{\pi_{\text{ref}}(y_c|x)} - \beta \log \frac{\pi(y_r|x)}{\pi_{\text{ref}}(y_r|x)})], \quad (1)$$

where $\sigma$ is the sigmoid function and $\beta$ is a hyperparameter that controls the proximity to the SFT model $\pi_{\text{ref}}$.

**Preference Dataset.** The performance of preference optimization largely depends on the choice of the preference dataset $\mathcal{D}_p$. Existing approaches explore the use of high-performance models (e.g., GPT-4) to synthesize high-quality instructions, responses, and preference feedback (Ding et al., 2023; Honovich et al., 2023; Cui et al., 2023; Mukherjee et al., 2023; Xu et al., 2024a; Liu et al., 2024a).

Several papers have investigated annotation-efficient learning by reducing the number of instructions rather than synthesizing more (Cohn et al., 1994; Settles, 2009). Su et al. (2023) suggested selecting examples to annotate from a pool of unlabeled data to improve the efficiency of in-context learning. Zhou et al. (2023) shows that fine-tuning a model with carefully selected and authored instructions can improve performance. Chen et al. (2024) points out that public instruction datasets contain many low-quality instances and proposes a method to filter out low-quality data, resulting in more efficient fine-tuning.

Regarding the selection of the response texts, several works have proposed to use the **West-of-N (WoN) strategy** (Xu et al., 2023; Yuan et al., 2024b; Pace et al., 2024). The WoN strategy randomly samples $N$ responses $\{y_i\}_{i=1}^N$ for each instruction $x$. Then, it annotates the preference *over all $N$ responses*. The response with the highest preference is labeled as chosen (win) $y_c$ and the one with the lowest preference is labeled as rejected (lose) $y_r$ to construct $\mathcal{D}_p$:

$$y_c \leftarrow \arg\max_{y\in\{y_i\}_{i=1}^N} R(x,y), \quad y_r \leftarrow \arg\min_{y\in\{y_i\}_{i=1}^N} R(x,y). \quad (2)$$

---

**Algorithm 1** Annotation-Efficient Preference Optimization (AEPO)

---

**Input:** A set of pairs of an instruction and a set of responses $\mathcal{D} = \{(x, Y_{\text{cand}})\}$, a preference
    annotator $R$, and an annotation budget per instruction $k$

  1: $\mathcal{D}_{AE} = \emptyset$
  2: **for** $(x, Y_{\text{cand}}) \in \mathcal{D}$ **do**
  3:     $Y^* \leftarrow \arg\max_{Y \subseteq Y_{\text{cand}}, |Y|=k} f_{rep}(Y) + \lambda f_{div}(Y)$     (See Eq. 18)
  4:     $y_c \leftarrow \arg\max_{y \in Y^*} R(x, y)$
  5:     $y_r \leftarrow \arg\min_{y \in Y^*} R(x, y)$
  6:     $\mathcal{D}_{AE} \leftarrow \mathcal{D}_{AE} \cup \{(x, y_c, y_r)\}$
  7: **end for**
  8: **return** $\mathcal{D}_{AE}$

---

The strategy is shown to be more efficient than random sampling with the same number of instructions. However, it requires $N$ annotations per instruction to run, making it inapplicable when the annotation budget is limited.

## 3    ANNOTATION-EFFICIENT PREFERENCE OPTIMIZATION (AEPO)

We propose **Annotation-Efficient Preference Optimization (AEPO)**, a method for efficiently learning preferences from a large number of responses *with a limited budget on preference annotations* (Figure 1).

The procedure of AEPO is described in Algorithm 1. We assume that a set of $N$ responses is available for each instruction: $\mathcal{D} = \{(x, \{y_i\}_{i=1}^{N})\}$. Instead of annotating the preference over all responses in $\{y_i\}_{i=1}^{N}$, AEPO subsamples $k$ responses (e.g., $k = 2$) from the candidate set of samples according to the objective function (Eq. 18) that heuristically maximizes the information gain (line 3). We explain the objective function later. Then, it deploys the WoN strategy (Eq. 2) on the subsampled subset of responses $Y^*$ instead of all $N$ responses $\{y_i\}_{i=1}^{N}$. It annotates the preference over $Y^*$ to select the best and the worst responses as the chosen and the rejected responses, respectively (lines 4, 5). In this way, we can allocate the annotation budget only to labeling informative responses. AEPO achieves to build a preference dataset with diverse and representative responses using a small amount of annotation effort, which is exactly the characteristics desired for the preference annotation methodology we discussed in Section 1.

The performance of the procedure is highly dependent on how we subsample a subset $Y$ from the candidate set of responses $Y_{\text{cand}} := \{y_i\}_{i=1}^{N}$. We propose to maximize the information gain (IG) (Cover, 1999) as the criteria to select the subset $Y$. Let $R^y$ be a random variable for the estimated probability distribution of $y$'s reward value ($R(x, y)$) and $R^Y$ be a set of random variables $R^y$ for $y \in Y$. The information gain $\text{IG}(R^{Y_{\text{cand}}}; R^Y)$ measures the reduction in the entropy of the predicted values of $R^{Y_{\text{cand}}}$ when we observe the values of $R^Y$:

$$\text{IG}(R^{Y_{\text{cand}}}; R^Y) = \mathbf{H}[R^{Y_{\text{cand}}}] - \mathbf{H}[R^{Y_{\text{cand}}} \mid R^Y], \tag{3}$$

where $\mathbf{H}$ is the joint entropy. Our goal is to find an informative subset $Y$ where $\text{IG}(R^{Y_{\text{cand}}}; R^Y)$ is maximized.

Information gain is one of the primary objectives used in active learning, where the goal is to selectively label the most informative unlabeled examples (Lewis & Gale, 1994; Engelson & Dagan, 1996; Guo & Greiner, 2007; Siddhant & Lipton, 2018; Nguyen et al., 2021; Huang et al., 2024a). We choose the subset $Y$ to label the preference so that the information gain for $R^{Y_{\text{cand}}}$ is maximized, which we assume will lead to better alignment.

Since the information gain is not computable in a feasible time for LLMs, we instead make two assumptions to heuristically estimate the information gain. Let $d$ be a cost function that represents the dissimilarity of the two response texts: $d : \mathcal{Y} \times \mathcal{Y} \rightarrow [0, 1]$, where $d(y, y') = 0$ if $y = y'$.

**Heuristic 1** *The preference annotation over $Y$ ($R^Y$) is more likely to be informative to $R^y$ if it is closer to $y$. That is, if*

$$\sum_{y_i \in Y} d(y, y_i) \leq \sum_{y_i \in Y'} d(y, y_i), \tag{4}$$

*then,*

$$\text{IG}(R^y; R^Y) \geq \text{IG}(R^y; R^{Y'}) \tag{5}$$

*with high probability.*

Figure 2 illustrates the intuition behind the heuristic. We assume that similar texts are more likely to have similar preferences. Thus, we assume that selecting a subset $Y$ closer to $y$ is more informative for estimating $R^y$ than a more distant subset $Y'$.

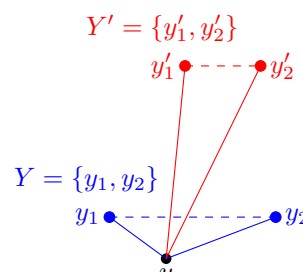

From Eq. 4, we are motivated to choose a subset $Y$ so that they are closer to $y \in Y_{\text{cand}}$:

$$f_{rep}(Y; y) := -\frac{1}{N} \sum_{y_i \in Y} d(y, y_i), \tag{6}$$

as a smaller $f_{rep}$ leads to larger expected information gain for $R^y$ (Eq. 5). Let $f_{rep}(Y)$ be the sum of $f_{rep}(y; Y)$ for $y \in Y_{\text{cand}}$:

$$f_{rep}(Y) := -\sum_{y \in Y_{\text{cand}}} f_{rep}(y; Y). \tag{7}$$

From the heuristic, the larger $f_{rep}(Y)$ is, the more likely it is that the information gain of $f_{rep}(Y)$ is greater.

Figure 2: An illustrative example of response subsets for annotating preference. Our algorithm is based on the heuristic that the subset $Y$ that is more diverse and closer to $y$ is more likely to be informative than $Y'$ to infer the value of $y$.

**Remark 1** *Assume Heuristic 1. The preference over $Y$ ($R^Y$) is more likely to be informative for estimating $R^{Y_{\text{cand}}}$ if it is closer to $Y_{\text{cand}}$. That is, If*

$$f_{rep}(Y) \geq f_{rep}(Y'), \tag{8}$$

*then*

$$\text{IG}(R^{Y_{\text{cand}}}; R^Y) \geq \text{IG}(R^{Y_{\text{cand}}}; R^{Y'}) \tag{9}$$

*with high probability.*

The remark is derived from the summation over $y \in Y_{\text{cand}}$ in Heuristic 1. As such, $f_{rep}(Y)$ is a reasonable objective to maximize the information gain (Eq. 3) under the given assumption.

An alternative explanation of $f_{rep}(Y)$ is that it quantifies the representativeness of the subset $Y$ for the entire sample set $Y_{\text{cand}}$.

$$f_{rep}(Y) = \sum_{y \in Y_{\text{cand}}} f_{rep}(y; Y) \tag{10}$$

$$= \sum_{y \in Y_{\text{cand}}} \left( -\frac{1}{N} \sum_{y \in Y} d(y, y') \right) \tag{11}$$

$$= -\sum_{y \in Y} \left( \frac{1}{N} \sum_{y' \in Y_{\text{cand}} \setminus \{y\}} d(y, y') \right) \tag{12}$$

where $\sum_{y' \in Y_{\text{cand}} \setminus \{y\}} d(y, y')$ can be interpreted as the average distance from $y$ to all other samples. That is, it shows the closeness to the mean of the sample set. Thus, the objective is to select a subset $Y$ that is closer to the center of the samples, making it more representative of the generated samples.

The second heuristic is about the effect of the diversity of a subset $Y$.

**Heuristic 2** *The preference over $Y$ ($R^Y$) is more likely to be informative for estimating $R^{Y_{\text{cand}}}$ if each pair of samples in $Y$ is more distinct. That is, if*

$$\sum_{y_1 \in Y} \sum_{y_2 \in Y \setminus \{y_1\}} d(y_1, y_2) \geq \sum_{y_1 \in Y'} \sum_{y_2 \in Y' \setminus \{y_1\}} d(y_1, y_2), \tag{13}$$

*then,*

$$\text{IG}(R^{Y_{\text{cand}}}; R^Y) \geq \text{IG}(R^{Y_{\text{cand}}}; R^{Y'}) \tag{14}$$

*with high probability.*

An example of high and low diversity subsamples ($Y$ and $Y'$) is shown in Figure 2. If the selected samples are too similar (e.g., $Y'$), then it will be difficult to infer $R^y$ when $y$ is different from both of them. On the other hand, if the selected samples are distinct enough (e.g., $Y$), then we expect it to be easier to infer $R^y$.

Motivated by the heuristic, we propose the following objective function $f_{div}$ as the diversity objective:

$$f_{div}(Y) = \frac{1}{|Y|} \sum_{y_1 \in Y} \sum_{y_2 \in Y \setminus \{y_1\}} d(y_1, y_2). \tag{15}$$

The objective $f_{div}(Y)$ is equal to the value of Eq. 13, so maximizing it improves the information gain to $R^{Y_{\text{cand}}}$.

An alternative view of $f_{div}$ is that it serves as an upper bound on the difference in distance to a pair of samples in $Y$, under the assumption that $d$ is a metric. Let $y_1, y_2$ be a pair of samples in $Y$ with $R(x, y_1) > R(x, y_2)$. It is difficult to infer $R^y$ when $|d(y, y_1) - d(y, y_2)|$ is small, since $y$ is roughly as close to $y_1$ as it is as to $y_2$ (Figure 2). Here, $d(y_1, y_2)$ is an upper bound of $|d(y, y_1) - d(y, y_2)|$ from the triangle inequality:

$$\forall y \ |d(y, y_1) - d(y, y_2)| \leq d(y_1, y_2). \tag{16}$$

Thus, $f_{div}(Y)$ serves as an upper bound on the sum of the difference in distance to a pair of subsampled texts $y_1$ and $y_2$:

**Remark 2** *Assume Heuristic 2. Let $d$ be a metric over $\mathcal{Y}$. $f_{div}$ is an upper bound on the sum of the distance difference between the sample pairs in $Y$.:*

$$\frac{1}{|Y|} \sum_{y \in Y_{\text{cand}}} \sum_{y_1 \in Y} \sum_{y_2 \in Y \setminus \{y_1\}} |d(y, y_1) - d(y, y_2)| \leq f_{div}(Y). \tag{17}$$

The proof is immediate from Eq. 16. Thus, it is ideal to have $f_{div}$ large enough so that $|d(y, y_1) - d(y, y_2)|$ is not too small to infer $R^{Y_{\text{cand}}}$. Although the cost functions used in NLP are often not metric (e.g., cosine distance), the remark serves as an intuitive explanation of the diversity objective $f_{div}$.

Based on the two heuristics, we propose to optimize the following objective to maximize the expected information gain from the subsample $Y$:

$$Y_k^* := \underset{\substack{Y \subseteq Y_{\text{cand}} \\ |Y| = k}}{\arg \max} f_{rep}(Y) + \lambda f_{div}(Y)$$

$$= \underset{\substack{Y \subseteq Y_{\text{cand}} \\ |Y| = k}}{\arg \max} - \sum_{y \in Y} \left( \frac{1}{N} \sum_{y' \in Y_{\text{cand}} \setminus \{y\}} d(y, y') \right) + \lambda \frac{1}{|Y|} \sum_{y_1 \in Y} \sum_{y_2 \in Y \setminus \{y_1\}} d(y_1, y_2), \tag{18}$$

where $\lambda$ is a hyperparameter to control the trade-off between the two objectives. We use the cosine distance of the embedding as the dissimilarity function:

$$d(y_1, y_2) = 1 - \cos(\text{emb}(y_1), \text{emb}(y_2)), \tag{19}$$

where $\cos$ is the cosine function and $\text{emb}$ is the embedding function. We use the `all-mpnet-base-v2` sentence BERT model as the embedding model because it has been shown to be effective for a variety of sentence embedding tasks (Reimers & Gurevych, 2019; 2020; Song et al., 2020).

## 4 EXPERIMENTS

**Setup.** We evaluate the performance of AEPO on DPO using the AlpacaFarm (Dubois et al., 2023) and Anthropic's hh-rlhf (Bai et al., 2022) datasets. We use mistral-7b-sft-beta (Mistral) (Jiang et al., 2023a; Tunstall et al., 2024) as the language model. See D.2 for the results using dolly-v2-3b (Conover et al., 2023) as the language model.

We generate up to $N = 128$ responses per instruction with nucleus sampling ($p = 0.9$) (Holtzman et al., 2020) to be used for the subsampling strategies. The temperature of the sampling algorithm is set to 1.0 for all experiments. All the methods use the same set of responses to ensure a fair comparison. For AEPO, the number of subsampled responses is set to $k = 2$ and the diversity hyperparameter is set to $\lambda \in \{0.0, 0.5, 1.0, 2.0\}$ for AlpacaFarm and $\lambda \in \{0.5, 1.0, 2.0\}$ for the rest of the datasets. We evaluate random sampling and WoN strategy as baselines. We additionally evaluate a coreset-based subsampling strategy (Sener & Savarese, 2018) and a perplexity-based subsampling strategy for AlpacaFarm. See Appendix B for the details of the algorithms. Since WoN strategy uses $N/2$ times more annotations per instruction than AEPO with $k = 2$, we reduce the number of instructions for WoN to $2/N$ so that the number of required annotations is the same as for AEPO. Note that we assume that the cost of annotating the preference rank for $N$ responses is linear in $N$. This assumption favors WoN because it becomes increasingly difficult to annotate preference rank over a larger set of options (Ganzfried, 2017).

We use the OASST reward model (Köpf et al., 2023) to annotate the preference over the responses for the training data. Although it is ideal to use human annotations to evaluate the performance of the algorithms, human annotations are expensive and difficult to reproduce. To this end, we use existing open source reward models as preference annotators for the experiment.

We train the same model that generates the responses (Mistral) using DPO with Low-Rank Adaptation (LoRA) (Hu et al., 2022; Sidahmed et al., 2024). We set the LoRA's $r = 64$ and $\alpha = r/4$. Other hyperparameters for the training process are described in Appendix A. For the Alpaca-Farm dataset, we use the `alpaca_human_preference` subset as the training set and use the `alpaca_farm_evaluation` subset as the evaluation set. For the Anthropic's hh-rlhf datasets, we use the first 5000 entries of the training set of both the `helpful-base` and `harmless-base` subsets as the training set. Then we evaluate the trained model on the first 1000 entries of the test set of the `helpful-base` (Helpfulness) and `harmless-base` (Harmlessness) subsets. For WoN, we reduce the number of instructions evenly for the two subsets so that the dataset always has the same number of instructions from the two subsets.

We evaluate the quality of the trained models by sampling a response using nucleus sampling ($p = 0.7$). The model output is evaluated using Eurus-RM-7B (Eurus) (Yuan et al., 2024a) as it is open source and shown to have a high correlation with human annotations in RewardBench (Lambert et al., 2024).

**Main Results.** Figure 3 shows the Eurus score of the DPO models on AlpacaFarm using AEPO ($\lambda = 1.0$) and WoN with different numbers of responses. WoN with $N = 4$ outperforms the random sampling baselines (i.e., WoN with $N = 2$), even though it uses only half of the available instructions, which is consistent with the results of Song et al. (2024). However, WoN's score drops significantly for $N \geq 8$ as the number of instructions decreases. In contrast, AEPO scales with the number of responses $N$ and outperforms WoN (Figure 3).

Figures 5 and 6 show the win rate of the DPO models with $N = 128$ under a fixed annotation budget. The win rate is computed against the SFT model using Eurus as a reference reward model. See Appendix H for the evaluation using other reward models. In all three datasets, AEPO outperforms the baseline algorithms except for when $\lambda$ is set to 0 so that no diversity is assured.

The ablation study of AEPO is described in Appendix D where we evaluate AEPO on a smaller LLM, out-of-domain tasks, using varying LoRA hyperparameters, and using varying loss functions. The result shows that AEPO consistently outperforms the baselines in a wide range of settings.

**AEPO generates a diverse and representative preference dataset.** We evaluate the diversity, representativeness, and quality of the preference dataset generated by AEPO with $k = 2$, $N \in \{2 \text{ (Random)}, 4, 8, 16, 32, 64, 128\}$, and $\lambda \in \{0, 0.3, 0.5, 1.0, 2.0\}$. To measure the semantic and

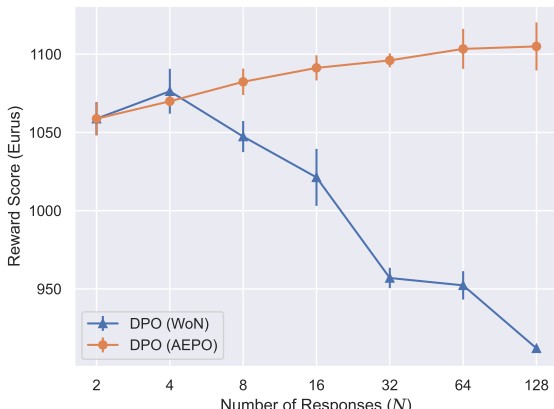

Figure 4: The number of instructions (#Insts) and annotations (#Annots) used by the preference annotation strategies in Figures 5, 6, and 8.

| Method | #Insts | #Annots |
|---|---|---|
| SFT (Mistral) | 0 | 0 |
| Random ($p = 0.8$) | $|\mathcal{D}|$ | $2|\mathcal{D}|$ |
| Random ($p = 0.9$) | $|\mathcal{D}|$ | $2|\mathcal{D}|$ |
| Random ($p = 1.0$) | $|\mathcal{D}|$ | $2|\mathcal{D}|$ |
| WoN ($N = 4$) | $|\mathcal{D}|/2$ | $2|\mathcal{D}|$ |
| WoN ($N = 8$) | $|\mathcal{D}|/4$ | $2|\mathcal{D}|$ |
| WoN ($N = 128$) | $|\mathcal{D}|/64$ | $2|\mathcal{D}|$ |
| Coreset | $|\mathcal{D}|$ | $2|\mathcal{D}|$ |
| Perplexity | $|\mathcal{D}|$ | $2|\mathcal{D}|$ |
| AEPO ($\lambda = 0$) | $|\mathcal{D}|$ | $2|\mathcal{D}|$ |
| AEPO ($\lambda = 0.5$) | $|\mathcal{D}|$ | $2|\mathcal{D}|$ |
| AEPO ($\lambda = 1.0$) | $|\mathcal{D}|$ | $2|\mathcal{D}|$ |
| AEPO ($\lambda = 2.0$) | $|\mathcal{D}|$ | $2|\mathcal{D}|$ |

Figure 3: Evaluation of AEPO and West-of-N for DPO *with an annotation budget fixed to* 2 *times the number of instructions* on AlpacaFarm. The line represents the average reward score and the bar shows the standard deviation over three runs.

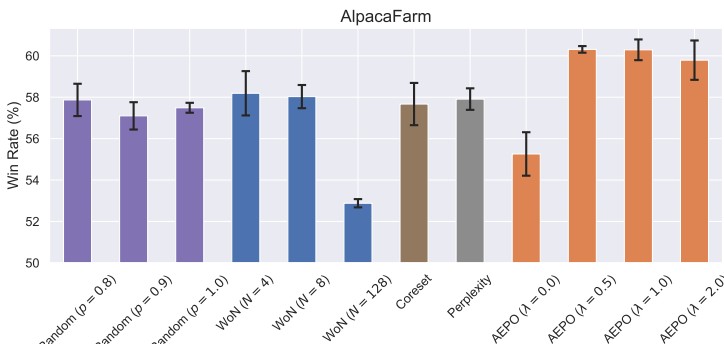

Figure 5: Evaluation of preference annotation strategies for DPO on AlpacaFarm using Mistral under the annotation budget fixed to 2 times the number of instructions. The win rate against the SFT model is evaluated. The bar represents the mean, and the error bar indicates the standard deviation of three runs.

lexical diversity of the responses, we use pairwise Sentence BERT and distinct-n (Li et al., 2016). We use the same Sentence BERT model (`all-mpnet-base-v2`) as AEPO to evaluate the average cosine similarity between the selected pairs of responses. Distinct-n counts the number of distinct n-grams in a sentence divided by the length of the sentence. The representativeness is measured by $-f_{rep}(Y)/|Y_{\text{cand}}|$ which is the average similarity ($-d(y, y')$) of the selected texts $Y$ to the whole sample set $Y_{\text{cand}}$. The quality of the responses is measured by the average reward score of the selected responses.

The result is shown in Figure 7a. By using a larger number of responses $N$, AEPO manages to generate more diverse and representative response pairs than a random sampling with the same number of annotations. Interestingly, AEPO also results in higher-quality texts being selected than random sampling (Figure 7b). This aligns with prior work reporting that diversity and representativeness objectives may also improve the quality of the output texts (Vijayakumar et al., 2016; 2018; Eikema & Aziz, 2022; Jinnai et al., 2024). See Appendix E for examples of the preference data generated by AEPO. We observe similar trends in the results on distinct-n, as well as the results on the Anthropic's datasets (Figures 15, 16, and 17 (Appendix H).

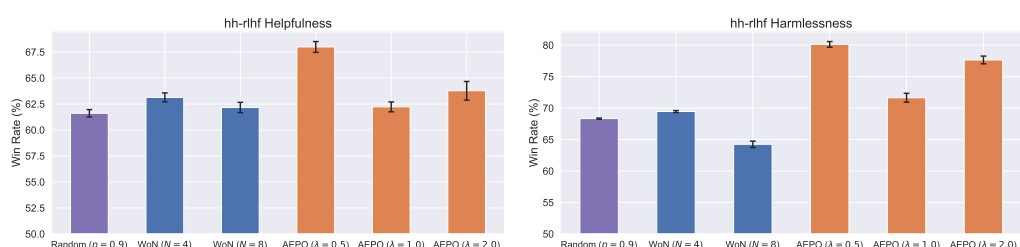

Figure 6: Evaluation of preference dataset annotation strategies for DPO on hh-rlhf's Helpfulness and Harmlessness dataset using Mistral under the annotation budget fixed to 2 times the number of instructions. The win rate against the SFT model is evaluated. The bar represents the mean, and the error bar indicates the standard deviation of three runs.

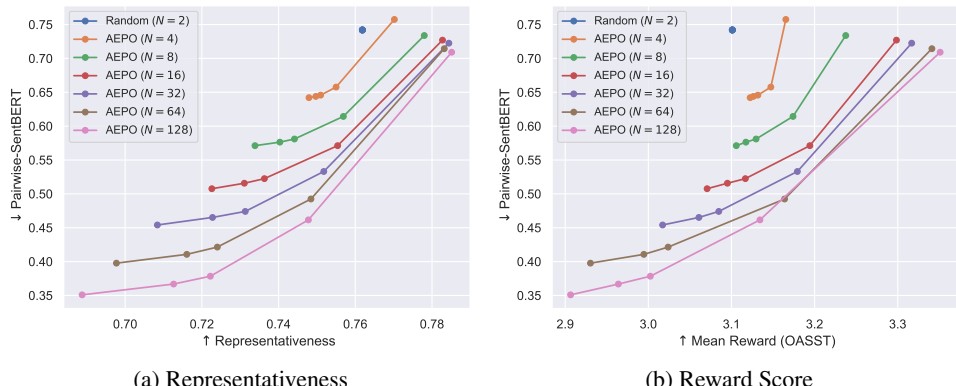

(a) Representativeness          (b) Reward Score

Figure 7: Diversity (↓Sentence BERT), representativeness, and quality (↑mean reward) of the responses of the preference datasets $\mathcal{D}_{AE}$ generated by the subsampling process of AEPO with a varying number of input responses ($N$). The number of selected responses ($k$) is fixed at 2. AEPO successfully generates datasets with better diversity-representativeness trade-offs and diversity-quality trade-offs without requiring additional annotations.

**Both diversity and representativeness of the preference dataset are important for preference learning.** The question is what contributes to the improved performance of AEPO. We evaluate AEPO with $\lambda \in \{0.0, 0.5, 1.0, 2.0\}$ to investigate the importance of diversity and representativeness of responses on AlpacaFarm dataset. AEPO with moderate size of $\lambda$ outperforms AEPO with higher or lower $\lambda$ (Figure 5 and 10). The result shows that both the diversity and the representativeness of responses are important for the preference dataset, which is consistent with the observations in previous work (Mukherjee et al., 2023; Chen et al., 2024; Liu et al., 2024c; Song et al., 2024).

**AEPO is effective for learning Japanese commonsense morality with a limited annotation budget.** To evaluate the proposed method in an application where the annotation budget is often limited, we conduct an experiment using the JCommonsenseMorality (JCM) dataset (Takeshita et al., 2023). JCM is a collection of texts labeled with whether a text contains a morally wrong statement according to the commonsense morality (Hendrycks et al., 2021) of people in Japanese culture. Because commonsense morality is culturally dependent and requires annotation by the members of the community (Durmus et al., 2024; Shen et al., 2024a), it is difficult to collect a large number of annotations. Therefore, we consider the task of learning Japanese commonsense morality to be a good benchmark for evaluating AEPO in a realistic application where AEPO is needed.

We use 800 entries ($|\mathcal{D}| = 800$) from the train split for training and 500 entries from the test split for evaluation. We train a Japanese LLM (`calm2-7b-chat`) using the train set of the JCM dataset (Sugimoto, 2024). As a reward model, we evaluate the accuracy of the output with respect to the

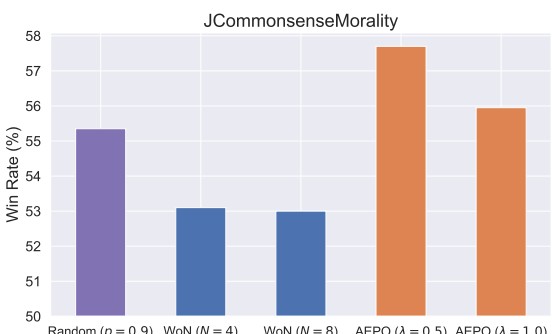

Figure 8: Evaluation of preference annotation strategies for DPO on the JCommonsenseMorality (JCM) dataset using `calm2-7b-chat` under a fixed annotation budget. The win rate against the SFT model is evaluated.

label provided in the dataset, as well as the overall quality. See Appendix G for the evaluation procedure. The results are summarized in Figure 8. Overall, AEPO outperforms the baselines within the same annotation budget constraint. The result on the JCM dataset suggests that AEPO is an effective strategy in one of the tasks where the available annotations are limited.

## 5 RELATED WORK

**Minimum Bayes risk decoding.** Eq. 7 and 18 are largely inspired by Minimum Bayes Risk (MBR) decoding (Kumar & Byrne, 2002; 2004; Eikema & Aziz, 2022). MBR decoding is a text generation algorithm that selects the sequence with the highest similarity to the sequences generated by the probability model. As such, the objective function of MBR decoding corresponds to Eq. 7. MBR decoding has been proven to produce high-quality text in many text generation tasks, including machine translation, text summarization, and image captioning (Freitag et al., 2023; Suzgun et al., 2023; Bertsch et al., 2023; Li et al., 2024a; Yang et al., 2024). In particular, Eq. 18 is strongly inspired by the objective function of Diverse MBR (DMBR) decoding (Jinnai et al., 2024). The novelty of our work is to introduce the objective function of DMBR as a strategy to subsample representative and diverse responses from many candidate responses so that the annotation budget can be used efficiently.

**Active learning.** Related work in active learning is described in Appendix C.

## 6 CONCLUSIONS

We propose Annotation-Efficient Preference Optimization (AEPO), an annotation-efficient dataset subsampling strategy for language model alignment. The subsampling strategy aims to maximize the information gain using two heuristics on how the preference information is propagated between samples. By focusing the annotation effort on the selected responses, AEPO achieves efficient preference optimization with a limited annotation budget. We evaluate the subsampling strategy and show that it successfully selects diverse and representative samples from the candidates (Figure 7). Experimental results show that AEPO outperforms the baselines on AlpacaFarm, Anthropic's hh-rlhf, and JCM datasets (Figures 5, 6, and 8). Our ablation study covers various settings, including GPT-4 evaluation, off-policy training, out-of-domain evaluation, and using different hyperparameters (Appendix D). The study shows that AEPO consistently outperforms the baselines in various settings. We believe that AEPO is a critical contribution to promoting preference optimization research by addressing the severe obstacle, the cost of creating better preference data.

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

## A  HYPERPARAMETERS

Table 2 lists the hyperparameters we use to run DPO. Table 3 lists the hyperparameters we use to generate the texts for evaluation.

Table 2: DPO hyperparameters.

| Parameter | Value |
|---|---|
| Training epochs | 3 |
| Batch size | 4 |
| Regularization factor ($\beta$) | 0.1 |
| Optimizer | RMSProp |
| Learning rate | 1e-5 |
| Learning rate scheduler | linear |
| Warm up steps | #instructions / 80 |
| Max instruction length | 512 |
| Max new tokens | 512 |
| Max total length | 512 |

Table 3: Generation hyperparameters on evaluation.

| Parameter | Value |
|---|---|
| Max instruction length | 512 |
| Max new tokens | 512 |
| Temperature | 1.0 |
| Top-$p$ | 0.7 |

## B  IMPLEMENTATION OF BASELINES

In addition to the existing methods (random sampling and WoN sampling), we present two response texts subsampling strategies, a coreset-based subsampling and perplexity-based subsampling as baselines.

We implement the Coreset selection using the set cover minimization algorithm following the work of Sener & Savarese (2018) (Algorithm 1, k-Center-Greedy). The objective function for selecting the subset $Y$ is the following:

$$Y^* = \arg\min_{Y \subseteq Y_{\text{cand}}} \max_{y \in Y_{\text{cand}}} \min_{y' \in Y} d(y, y'). \tag{20}$$

Intuitively, Eq. 20 is similar to the representative objective ($f_{rep}$; Eq. 7) but instead of minimizing the average distance of $Y$ and $Y_{\text{cand}}$, it aims to minimize the maximum distance of $y \in Y_{\text{cand}}$ and $y' \in Y$. Although the algorithm was originally proposed for training convolutional neural networks, its procedure applies to the response text subsampling problem. We use the cosine distance of the sentence embedding as the distance between the data points. We use the same text embedding model as AEPO (`all-mpnet-base-v2`).

The perplexity-based dataset filtering strategy is shown to be effective for the pretraining (De la Rosa et al., 2022; Marion et al., 2023; Thrush et al., 2024) and instruction fine-tuning (Zhou et al., 2023; Li et al., 2024b). We implement a perplexity-based selection strategy to pick a pair of responses with the highest and the lowest perplexity:

$$Y^* = \{\arg\max_{y \in Y_{\text{cand}}} PP(y \mid x), \arg\min_{y \in Y_{\text{cand}}} PP(y \mid x)\}, \tag{21}$$

where $PP$ denotes the perplexity of $y$ given $x$ as the input.

## C    ADDITIONAL RELATED WORK

**Active learning.**    Annotation-efficient learning has long been a challenge in natural language processing (Zhang et al., 2022). Active learning is an approach that aims to achieve training with fewer training labels by proactively selecting the data to be annotated and used for learning (Cohn et al., 1994; Settles, 2009; Houlsby et al., 2011). There are roughly two active learning strategies used in NLP (Zhang et al., 2022). One uses the informativeness of the data instances, such as uncertainty and disagreement of the models (Lewis & Gale, 1994; Engelson & Dagan, 1996; Siddhant & Lipton, 2018; Huang et al., 2024a). This approach has proven to be efficient in many text classification tasks. The other strategy is based on the representativeness of the data instances (McCallum & Nigam, 1998; Settles & Craven, 2008; Zhao et al., 2020; Chen & Wang, 2024). The strategy annotates instances with high average similarity to all the other instances so that it can cover a large portion of the dataset with few annotations. Another approach is to select instances that maximize the diversity of labeled instances (Eck et al., 2005; Zeng et al., 2019; Bloodgood & Callison-Burch, 2010). Our approach is related to these approaches as our objective is a combination of representative and diversity measures designed to maximize the information gain. The novelty of our study lies in applying these ideas to the language model alignment problem to reduce the annotation cost.

## D    ABLATION STUDY

We describe the ablation study to evaluate the effect of AEPO in various settings.

### D.1    GPT-4 EVALUATION

Figure 9 shows the win rate of the DPO models against the SFT model using GPT-4 as an evaluator. Overall we observe the same qualitative result as in Eurus. We access GPT-4 API via Azure OpenAI service. The model name is gpt-4o and the model version is 2024-05-13. We set the model temperature, frequency penalty, and presence penalty to 0. The following prompt is used to evaluate the response text:

> Please act as an impartial judge and evaluate the quality of the response provided by an AI assistant to the user question displayed below. Your evaluation should consider factors such as the helpfulness, relevance, accuracy, depth, creativity, and level of detail of the response. Begin your evaluation by providing a short explanation. Be as objective as possible. After providing your explanation, you must rate the response on a scale of 1 to 10 by strictly following this format: "[[rating]]", for example: "Rating: [[5]]".
>
> [Question]
> {question}
> [The Start of Assistant's Answer]
> {answer}
> [The End of Assistant's Answer]

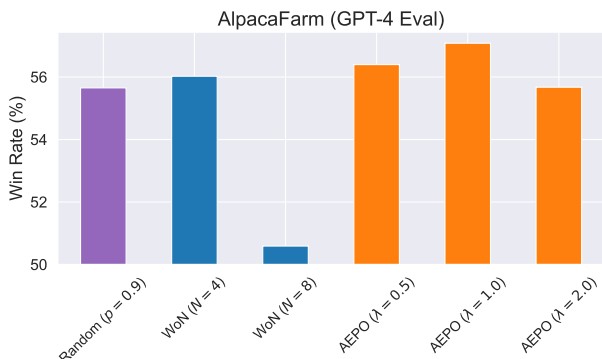

Figure 9: Evaluation of AEPO on the AlpacaFarm dataset using GPT-4 as an evaluator. The win rate against the SFT model is evaluated.

## D.2 TRAINING DOLLY LANGUAGE MODEL

Several studies have shown that using responses generated by the training model itself (on-policy learning) is more effective than using responses generated by other models (off-policy learning) (Chang et al., 2024; Guo et al., 2024; Xu et al., 2024c; Tajwar et al., 2024; Dong et al., 2024; Pace et al., 2024; Tang et al., 2024a). Nevertheless, off-policy learning is advantageous in resource-constrained settings because it can leverage existing public resources to train arbitrary models.

To this end, we investigate the use of AEPO for off-policy learning. We use the preference dataset $\mathcal{D}_{AE}$ generated by Mistral's responses $\{y_i\}_{i=1}^N$ on AlpacaFarm to train dolly-v2-3b (Dolly; Conover et al. 2023). We set the LoRA's $r = 32$ and $\alpha = r/4$. Other experimental settings are the same as the experiment on Mistral. Figure 10 shows the results of the off-policy learning using Eurus as the reference reward model. AEPO with sufficiently large $\lambda$ outperforms vanilla DPO. The result shows the potential of AEPO to improve the efficiency of off-policy learning. See Table 18 for the result using other reward models.

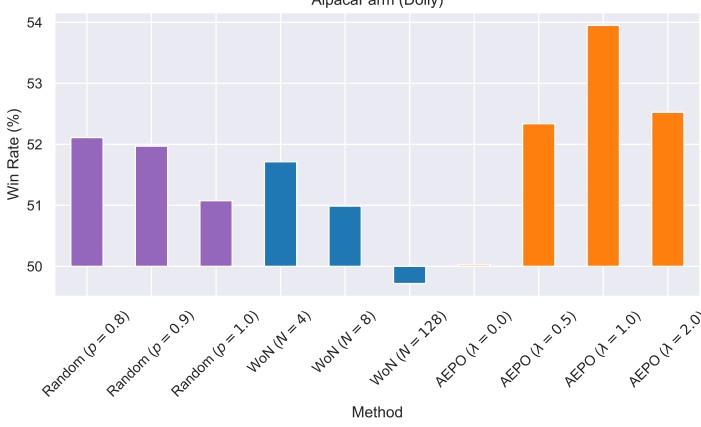

Figure 10: Evaluation of AEPO on training Dolly language model using the AlpacaFarm dataset. We generate responses with Mistral and use the sampled responses to train Dolly. The win rate against the SFT model is evaluated.

## D.3 OUT-OF-DOMAIN EVALUATION

Previous work has shown that training on a diverse set of instructions improves the performance on out-of-domain tasks (Sanh et al., 2022). The question is whether we can achieve a similar robustness

with a diverse set of responses generated by AEPO. We evaluate the Mistral models fine-tuned with the AlpacaFarm dataset on ARC (Clark et al., 2018), HellaSwag (Zellers et al., 2019), TruthfulQA (Lin et al., 2022), and WinoGrande (Sakaguchi et al., 2021) using the language model evaluation harness (Gao et al., 2023b). Table 4 summarizes the scores and the standard errors of the trained models on these benchmarks. Overall, AEPO scores slightly higher than WoN, except for the ARC. The result shows that AEPO outperforms WoN in the AlpacaFarm domain not because it overfits to the task, but because it improves on a wide range of tasks.

Table 4: Evaluation of DPO models trained with AlpacaFarm on out-of-domain benchmarks. Means and standard errors are reported.

| Preference Dataset Configuration | | | | | | |
|---|---|---|---|---|---|---|
| Method | #Insts | #Annots | ARC | HellaSwag | TruthfulQA | WinoGrande |
| SFT (Mistral) | 0 | 0 | $57.94 \pm 1.44$ | $82.07 \pm 0.38$ | $42.98 \pm 1.46$ | $77.51 \pm 1.17$ |
| Random ($p = 0.9$) | $|\mathcal{D}|$ | $2|\mathcal{D}|$ | $59.73 \pm 1.43$ | $83.14 \pm 0.37$ | $46.37 \pm 1.51$ | $78.06 \pm 1.16$ |
| WoN ($N = 4$) | $|\mathcal{D}|/2$ | $2|\mathcal{D}|$ | $59.73 \pm 1.43$ | $82.95 \pm 0.38$ | $48.13 \pm 1.54$ | $75.14 \pm 1.21$ |
| WoN ($N = 8$) | $|\mathcal{D}|/4$ | $2|\mathcal{D}|$ | $\mathbf{59.90} \pm 1.43$ | $82.80 \pm 0.38$ | $49.41 \pm 1.55$ | $74.90 \pm 1.22$ |
| AEPO ($\lambda = 0$) | $|\mathcal{D}|$ | $2|\mathcal{D}|$ | $59.64 \pm 1.43$ | $83.10 \pm 0.37$ | $46.31 \pm 1.51$ | $\mathbf{78.14} \pm 1.16$ |
| AEPO ($\lambda = 0.5$) | $|\mathcal{D}|$ | $2|\mathcal{D}|$ | $\mathbf{59.90} \pm 1.43$ | $\mathbf{83.28} \pm 0.37$ | $\mathbf{49.69} \pm 1.54$ | $77.19 \pm 1.18$ |
| AEPO ($\lambda = 1.0$) | $|\mathcal{D}|$ | $2|\mathcal{D}|$ | $58.62 \pm 1.44$ | $82.57 \pm 0.38$ | $44.34 \pm 1.49$ | $77.90 \pm 1.17$ |
| AEPO ($\lambda = 2.0$) | $|\mathcal{D}|$ | $2|\mathcal{D}|$ | $58.70 \pm 1.44$ | $82.54 \pm 0.38$ | $44.75 \pm 1.49$ | $77.58 \pm 1.17$ |

### D.4 LoRA Hyperparameters

We evaluate the effect of the LoRA hyperparameters on the performance of AEPO. We run DPO once with LoRA's $r \in \{32, 128\}$ and $\alpha = r/4$. All other experimental settings are the same as in Section 4. Tables 5 and 6 show the experimental results. We observe that AEPO outperforms WoN in reward scores as in Section 4 regardless of the choice of the LoRA's $r$.

Table 5: Evaluation of AEPO on AlpacaFarm using Mistral with LoRA's $r = 32$ and $\alpha = r/4$.

| Preference Dataset Configuration | | | | | | | |
|---|---|---|---|---|---|---|---|
| Method | #Insts | #Annots | OASST | Eurus | OASST (w%) | Eurus (w%) | PairRM (w%) |
| SFT (Mistral) | 0 | 0 | 1.901 | 878.48 | 50 | 50 | 50 |
| Random ($p = 0.8$) | $|\mathcal{D}|$ | $2|\mathcal{D}|$ | 2.021 | 997.05 | 54.22 | 55.59 | 52.49 |
| Random ($p = 0.9$) | $|\mathcal{D}|$ | $2|\mathcal{D}|$ | 2.029 | 970.77 | 54.10 | 54.72 | 52.64 |
| Random ($p = 1.0$) | $|\mathcal{D}|$ | $2|\mathcal{D}|$ | 2.099 | 1009.53 | 55.47 | 56.96 | 53.64 |
| WoN ($N = 4$) | $|\mathcal{D}|/2$ | $2|\mathcal{D}|$ | 2.088 | 1031.62 | 56.34 | 56.71 | 53.98 |
| WoN ($N = 8$) | $|\mathcal{D}|/4$ | $2|\mathcal{D}|$ | 2.052 | 993.94 | 54.84 | 56.09 | 54.10 |
| AEPO ($\lambda = 0$) | $|\mathcal{D}|$ | $2|\mathcal{D}|$ | 1.994 | 936.94 | 53.48 | 53.35 | 53.10 |
| AEPO ($\lambda = 0.5$) | $|\mathcal{D}|$ | $2|\mathcal{D}|$ | 2.079 | 981.37 | 56.77 | 55.53 | $\mathbf{54.12}$ |
| AEPO ($\lambda = 1.0$) | $|\mathcal{D}|$ | $2|\mathcal{D}|$ | $\mathbf{2.121}$ | $\mathbf{1063.08}$ | $\mathbf{58.26}$ | $\mathbf{58.07}$ | 53.98 |
| AEPO ($\lambda = 2.0$) | $|\mathcal{D}|$ | $2|\mathcal{D}|$ | 2.072 | 1034.58 | 55.53 | 56.34 | 53.97 |
| WoN ($N = 128$) | $|\mathcal{D}|$ | $128|\mathcal{D}|$ | 2.339 | 1169.37 | 65.47 | 63.23 | 59.61 |

Table 6: Evaluation of AEPO on AlpacaFarm using Mistral with LoRA's $r = 128$ and $\alpha = r/4$.

| Method | #Insts | #Annots | OASST | Eurus | OASST (w%) | Eurus (w%) | PairRM (w%) |
|--------|--------|---------|-------|-------|------------|------------|-------------|
| | | | Preference Dataset Configuration | | | | |
| SFT (Mistral) | 0 | 0 | 1.901 | 878.48 | 50 | 50 | 50 |
| Random ($p = 0.8$) | $\|\mathcal{D}\|$ | $2\|\mathcal{D}\|$ | 2.310 | 1149.53 | 63.11 | 60.62 | 59.18 |
| Random ($p = 0.9$) | $\|\mathcal{D}\|$ | $2\|\mathcal{D}\|$ | 2.394 | 1140.02 | 65.96 | 59.25 | 60.00 |
| Random ($p = 1.0$) | $\|\mathcal{D}\|$ | $2\|\mathcal{D}\|$ | 2.308 | 1096.25 | 63.11 | 58.01 | 58.96 |
| WoN ($N = 4$) | $\|\mathcal{D}\|/2$ | $2\|\mathcal{D}\|$ | 2.390 | 1160.43 | 66.02 | 63.66 | 61.68 |
| WoN ($N = 8$) | $\|\mathcal{D}\|/4$ | $2\|\mathcal{D}\|$ | 2.357 | 1183.47 | 65.65 | 63.29 | 61.28 |
| AEPO ($\lambda = 0$) | $\|\mathcal{D}\|$ | $2\|\mathcal{D}\|$ | 2.186 | 1050.34 | 60.62 | 58.01 | 57.80 |
| AEPO ($\lambda = 0.5$) | $\|\mathcal{D}\|$ | $2\|\mathcal{D}\|$ | 2.379 | 1172.73 | 63.29 | **63.91** | 60.37 |
| AEPO ($\lambda = 1.0$) | $\|\mathcal{D}\|$ | $2\|\mathcal{D}\|$ | 2.354 | 1164.29 | 64.35 | 63.60 | **60.62** |
| AEPO ($\lambda = 2.0$) | $\|\mathcal{D}\|$ | $2\|\mathcal{D}\|$ | **2.400** | **1203.51** | **66.34** | 63.60 | 59.69 |
| WoN ($N = 128$) | $\|\mathcal{D}\|$ | $128\|\mathcal{D}\|$ | 2.705 | 1303.34 | 74.35 | 68.76 | 66.72 |

## D.5 Loss Function

Several variants of loss functions are proposed to replace the sigmoid loss function of DPO. The experimental results of AEPO using hinge loss (Zhao et al., 2023; Liu et al., 2024b) and KTO loss (Ethayarajh et al., 2024) are given in Tables 7 and 8. We use LoRA $r = 32$ and LoRA $\alpha = r/4$. Other experimental settings follow the settings in Section 4. We observe that AEPO outperforms the baselines regardless of the choice of the loss function.

Table 7: Evaluation of AEPO on AlpacaFarm with Mistral using hinge loss.

| Preference Dataset Configuration | | | | | | | |
|---|---|---|---|---|---|---|---|
| Method | #Insts | #Annots | OASST | Eurus | OASST (w%) | Eurus (w%) | PairRM (w%) |
| SFT (Mistral) | 0 | 0 | 1.901 | 878.48 | 50 | 50 | 50 |
| Random ($p = 0.8$) | $\|\mathcal{D}\|$ | $2\|\mathcal{D}\|$ | 2.026 | 998.26 | 54.66 | 55.78 | 52.77 |
| Random ($p = 0.9$) | $\|\mathcal{D}\|$ | $2\|\mathcal{D}\|$ | 2.036 | 989.09 | 55.47 | 55.71 | 53.32 |
| Random ($p = 1.0$) | $\|\mathcal{D}\|$ | $2\|\mathcal{D}\|$ | 2.068 | 997.99 | 55.59 | 56.46 | 53.46 |
| WoN ($N = 4$) | $\|\mathcal{D}\|/2$ | $2\|\mathcal{D}\|$ | _2.095_ | 1009.54 | 55.90 | 55.28 | 53.69 |
| WoN ($N = 8$) | $\|\mathcal{D}\|/4$ | $2\|\mathcal{D}\|$ | 2.037 | 989.60 | 54.47 | 55.59 | _54.15_ |
| AEPO ($\lambda = 0$) | $\|\mathcal{D}\|$ | $2\|\mathcal{D}\|$ | 1.994 | 964.50 | 53.48 | 54.60 | 53.10 |
| AEPO ($\lambda = 0.5$) | $\|\mathcal{D}\|$ | $2\|\mathcal{D}\|$ | 2.079 | 991.11 | _56.77_ | 55.65 | **54.22** |
| AEPO ($\lambda = 1.0$) | $\|\mathcal{D}\|$ | $2\|\mathcal{D}\|$ | **2.121** | **1052.23** | **58.26** | **58.51** | 53.98 |
| AEPO ($\lambda = 2.0$) | $\|\mathcal{D}\|$ | $2\|\mathcal{D}\|$ | 2.072 | _1050.30_ | 55.53 | _57.27_ | 53.97 |
| WoN ($N = 128$) | $\|\mathcal{D}\|$ | $128\|\mathcal{D}\|$ | 2.335 | 1156.37 | 63.42 | 63.17 | 59.08 |

Table 8: Evaluation of AEPO on AlpacaFarm with Mistral using KTO loss.

| Preference Dataset Configuration | | | | | | | |
|---|---|---|---|---|---|---|---|
| Method | #Insts | #Annots | OASST | Eurus | OASST (w%) | Eurus (w%) | PairRM (w%) |
| SFT (Mistral) | 0 | 0 | 1.901 | 878.48 | 50 | 50 | 50 |
| Random ($p = 0.8$) | $\|\mathcal{D}\|$ | $2\|\mathcal{D}\|$ | 2.025 | 1022.52 | 54.78 | 57.14 | 52.83 |
| Random ($p = 0.9$) | $\|\mathcal{D}\|$ | $2\|\mathcal{D}\|$ | 2.057 | 988.42 | 55.16 | 55.90 | 53.04 |
| Random ($p = 1.0$) | $\|\mathcal{D}\|$ | $2\|\mathcal{D}\|$ | _2.095_ | 1000.09 | 56.15 | 57.02 | 53.88 |
| WoN ($N = 4$) | $\|\mathcal{D}\|/2$ | $2\|\mathcal{D}\|$ | 2.075 | 994.79 | 55.22 | 54.60 | _54.03_ |
| WoN ($N = 8$) | $\|\mathcal{D}\|/4$ | $2\|\mathcal{D}\|$ | 2.032 | 1002.73 | 54.29 | 56.15 | 53.87 |
| AEPO ($\lambda = 0$) | $\|\mathcal{D}\|$ | $2\|\mathcal{D}\|$ | 1.994 | 952.70 | 53.48 | 53.73 | 53.10 |
| AEPO ($\lambda = 0.5$) | $\|\mathcal{D}\|$ | $2\|\mathcal{D}\|$ | 2.079 | 981.76 | _56.77_ | 55.40 | **54.15** |
| AEPO ($\lambda = 1.0$) | $\|\mathcal{D}\|$ | $2\|\mathcal{D}\|$ | **2.121** | **1059.95** | **58.26** | **58.76** | 53.98 |
| AEPO ($\lambda = 2.0$) | $\|\mathcal{D}\|$ | $2\|\mathcal{D}\|$ | 2.072 | _1030.45_ | 55.53 | _57.39_ | 53.97 |
| WoN ($N = 128$) | $\|\mathcal{D}\|$ | $128\|\mathcal{D}\|$ | 2.359 | 1158.02 | 64.84 | 62.86 | 59.84 |

# E  GENERATION EXAMPLES OF AEPO

We show examples of the dataset entries generated by AEPO and WoN in Tables 9 and 10. AEPO with $\lambda = 0.0$ selects two very similar or even identical responses. In contrast, the two responses selected by AEPO with $\lambda = 1.0$ are sufficiently diverse.

Table 9: Example of a preference dataset entry generated by AEPO and WoN on AlpacaFarm dataset. The input prompt is: *Write two sentences with the same meaning but with different wordings\n\nHe went to the store.*

| Method | Label | Text |
|---|---|---|
| Random ($p = 0.9$) | Chosen | He visited the store. He went shopping. |
| | Rejected | He visited the store.
He went shopping.

I got a message from my mom. |
| WoN ($N = 4$) | Chosen | 1. He went to the market.
2. He went shopping. |
| | Rejected | He purchased products from the grocery store.
He went and made purchases at the store.

He left the house.
He walked away from his living space. |
| WoN ($N = 8$) | Chosen | 1. He went to the market.
2. He went shopping. |
| | Rejected | He purchased products from the grocery store.
He went and made purchases at the store.

He left the house.
He walked away from his living space. |
| AEPO ($\lambda = 0.0$) | Chosen | He went shopping at the store.
He visited a store for purchasing goods. |
| | Rejected | He traveled to the store or he went to shop. |
| AEPO ($\lambda = 1.0$) | Chosen | 1. The subject went to the destination.
2. The subject made his/her way to the location in question. |
| | Rejected | He made a trip to the store for some supplies.

He purchased items from the store.

I will buy the supplies from the store tomorrow.

Could you please provide me with some groceries from the store?

Can you tell me if the store has any fresh fruits and vegetables available for sale? |
| AEPO ($\lambda = 2.0$) | Chosen | 1. The subject went to the destination.
2. The subject made his/her way to the location in question. |
| | Rejected | I went to the local shop. |

Table 10: Example of a preference dataset entry generated by AEPO and WoN on AlpacaFarm dataset. The input prompt is: *Paraphrase this sentence to have the same meaning.\n\nWe have had a prolonged period of extremely cold weather.*

| Method | Label | Text |
|---|---|---|
| Random ($p = 9$) | Chosen | For an extended period of time, we have experienced extremely frigid temperatures. |
| | Rejected | The cold weather has been ongoing for a long period of time. |
| WoN ($N = 4$) | Chosen | During an extended period of harsh winter conditions, we have been dealing with exceptionally low temperatures. |
| | Rejected | For a very long time, the temperature has been extremely low. |
| WoN ($N = 8$) | Chosen | There has been a long stretch of bitterly cold weather. OR Cold weather has persisted for an extended period of time. |
| | Rejected | For a long time, the weather has been very cold. |
| AEPO ($\lambda = 0.0$) | Chosen | For an extended period of time, the weather has been extremely cold. |
| | Rejected | For an extended period of time, the weather has been extremely cold. |
| AEPO ($\lambda = 1.0$) | Chosen | The weather has been exceptionally frigid for an extended time. |
| | Rejected | We have experienced a long time with very low temperatures. |
| AEPO ($\lambda = 2.0$) | Chosen | The period of extremely low temperatures has persisted for an extended period of time. |
| | Rejected | For a long time now, we have experienced a severe cold spell. |

# F    EVALUATION OF WEST-OF-N STRATEGY

WoN is an effective strategy when an abundance of annotations is available. Table 11 shows the performance of DPO with the WoN strategy using $N$ annotations per instruction without reducing the size of the instruction set. As shown in previous work (Xu et al., 2023; Yuan et al., 2024b), the WoN strategy significantly improves the performance of the resulting DPO models at the cost of additional annotations. The win rate against the SFT model is shown in Figure 11.

Table 11: Evaluation of DPO with the WoN strategy on AlpacaFarm using Mistral. The results of $N = 2, 128$ are the average of three runs, while the rest are of a single run.

| Preference Dataset Configuration | | | | | | | |
| Method | #Insts | #Annots | OASST | Eurus | OASST (w%) | Eurus (w%) | PairRM (w%) |
| --- | --- | --- | --- | --- | --- | --- | --- |
| SFT (Mistral) | 0 | 0 | 1.901 | 878.48 | 50 | 50 | 50 |
| Random ($p = 0.9$) | $|\mathcal{D}|$ | $2|\mathcal{D}|$ | 2.174 | 1058.78 | 59.71 | 57.10 | 55.54 |
| WoN ($N = 4$) | $|\mathcal{D}|$ | $4|\mathcal{D}|$ | 2.315 | 1105.60 | 64.35 | 61.37 | 59.26 |
| WoN ($N = 8$) | $|\mathcal{D}|$ | $8|\mathcal{D}|$ | 2.422 | 1225.22 | 66.09 | 67.20 | 62.73 |
| WoN ($N = 16$) | $|\mathcal{D}|$ | $16|\mathcal{D}|$ | 2.454 | 1237.81 | 68.14 | 64.66 | 63.42 |
| WoN ($N = 32$) | $|\mathcal{D}|$ | $32|\mathcal{D}|$ | 2.529 | 1246.53 | 70.56 | 66.09 | 64.51 |
| WoN ($N = 64$) | $|\mathcal{D}|$ | $64|\mathcal{D}|$ | 2.470 | 1248.19 | 68.07 | 66.58 | 63.72 |
| WoN ($N = 128$) | $|\mathcal{D}|$ | $128|\mathcal{D}|$ | 2.522 | 1262.77 | 70.12 | 67.31 | 64.17 |

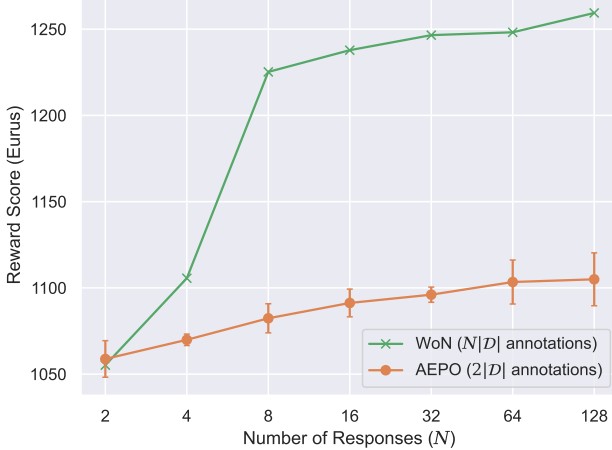

Figure 11: Evaluation of the DPO using the WoN strategy without limiting the annotation budget on AlpacaFarm using Mistral. The Eurus reward model is used for evaluation. With more annotations, WoN outperforms AEPO.

## G EVALUATION ON JCOMMONSENSEMORALITY

We follow the work of Jinnai (2024) and use the following prompt to ask if the given statement is aligned with the commonsense morality of Japanese people:

> 以下の文章に対してそれが道徳的あるいは倫理的な誤りがあるかを判断
> してください。 (*Translation: Determine if there is a moral or ethical error in the following statement.*)
>
> {question}

where we replace {question} with the statement in the dataset.

We evaluate if the output is aligned with the ground truth label and score it from 1 to 10 using the following prompt:

> Please act as an impartial judge and evaluate the quality of the response provided by an AI assistant to the user question displayed below. The correct answer of the problem is provided below. The response of the system has to be aligned with this correct answer. Your evaluation should consider factors such as the helpfulness, relevance, accuracy, depth, creativity, and level of detail of the response. Begin your evaluation by providing a short explanation. Be as objective as possible. After providing your explanation, you must rate the response on a scale of 1 to 10 by strictly following this format: "[[rating]]", for example: "Rating: [[5]]".
>
> [Question]
> 以下の文章に対してそれが道徳的あるいは倫理的な誤りがあるかを判断
> してください。 (*Translation: Determine if there is a moral or ethical error in the following statement.*)
> {question}
> [Correct Answer to the Question]
> {label}
> [The Start of Assistant's Answer]
> {answer}
> [The End of Assistant's Answer]

where we replace {question}, {label}, and {answer} with the question, the ground truth label, and the response of the model for each output. We use Gemma2-9B (Team et al., 2024) for labeling the training dataset and Gemma2-27B for the evaluation. All other experimental settings are the same as in Section 4.

## H ADDITIONAL RESULTS

### H.1 REWARD SCORES

Tables 12 and 13 show the results of training Mistral on AlpacaFarm including the results of the proxy reward model (OASST). The results of Mistral on Anthropic's Helpfulness and Harmlessness are shown in Tables 14, 15, 16, and 17. Table 18 is the result of training Dolly.

Interestingly, we observed that AEPO outperforms WoN with 64 times more annotations in Anthropic's datasets (Tables 14, 15, 16, and 17). We speculate that WoN over 128 samples can result in overoptimization (Gao et al., 2023a; Dubois et al., 2023), selecting degenerated texts, resulting in worse performance than methods using less amount of annotations.

Table 12: Reward score of the AEPO on AlpacaFarm using Mistral. The best score is in bold, and the second best is underlined. The mean and standard deviation of three runs are shown. Note that OASST is used as a proxy reward model to annotate the preference of the training dataset.

| Preference Dataset Configuration | | | | |
|---|---|---|---|---|
| Method | #Insts | #Annots | OASST | Eurus |
| SFT (Mistral) | 0 | 0 | 1.901 | 878.48 |
| Random ($p = 0.8$) | $\|\mathcal{D}\|$ | $2\|\mathcal{D}\|$ | $2.155 \pm 0.010$ | $1088.71 \pm 17.90$ |
| Random ($p = 0.9$) | $\|\mathcal{D}\|$ | $2\|\mathcal{D}\|$ | $2.174 \pm 0.009$ | $1058.78 \pm 10.60$ |
| Random ($p = 1.0$) | $\|\mathcal{D}\|$ | $2\|\mathcal{D}\|$ | $2.168 \pm 0.007$ | $1044.35 \pm 0.98$ |
| WoN ($N = 4$) | $\|\mathcal{D}\|/2$ | $2\|\mathcal{D}\|$ | $2.217 \pm 0.012$ | $1076.31 \pm 14.35$ |
| WoN ($N = 8$) | $\|\mathcal{D}\|/4$ | $2\|\mathcal{D}\|$ | $2.197 \pm 0.005$ | $1047.37 \pm 9.94$ |
| WoN ($N = 128$) | $\|\mathcal{D}\|/64$ | $2\|\mathcal{D}\|$ | $1.926 \pm 0.005$ | $912.03 \pm 1.25$ |
| Coreset | $\|\mathcal{D}\|$ | $2\|\mathcal{D}\|$ | $2.107 \pm 0.011$ | $1037.100 \pm 11.31$ |
| Perplexity | $\|\mathcal{D}\|$ | $2\|\mathcal{D}\|$ | $2.187 \pm 0.008$ | $1051.52 \pm 15.54$ |
| AEPO ($\lambda = 0$) | $\|\mathcal{D}\|$ | $2\|\mathcal{D}\|$ | $2.063 \pm 0.009$ | $999.03 \pm 1.43$ |
| AEPO ($\lambda = 0.5$) | $\|\mathcal{D}\|$ | $2\|\mathcal{D}\|$ | $\mathbf{2.230} \pm 0.011$ | $\underline{1094.20} \pm 13.70$ |
| AEPO ($\lambda = 1.0$) | $\|\mathcal{D}\|$ | $2\|\mathcal{D}\|$ | $\underline{2.222} \pm 0.009$ | $\mathbf{1104.97} \pm 15.33$ |
| AEPO ($\lambda = 2.0$) | $\|\mathcal{D}\|$ | $2\|\mathcal{D}\|$ | $2.219 \pm 0.010$ | $1085.78 \pm 9.72$ |
| WoN ($N = 128$) | $\|\mathcal{D}\|$ | $128\|\mathcal{D}\|$ | $2.522 \pm 0.008$ | $1262.77 \pm 5.62$ |

Table 13: Win rate against the SFT model (Mistral) on AlpacaFarm. The best score is in bold, and the second best is underlined. The mean and standard deviation of three runs are shown. Note that OASST is used as a proxy reward model to annotate the preference of the training dataset.

| Preference Dataset Configuration | | | | | |
|---|---|---|---|---|---|
| Method | #Insts | #Annots | OASST (w%) | Eurus (w%) | PairRM (w%) |
| SFT (Mistral) | 0 | 0 | 50 | 50 | 50 |
| Random ($p = 0.8$) | $\|\mathcal{D}\|$ | $2\|\mathcal{D}\|$ | $59.86 \pm 1.44$ | $57.87 \pm 0.78$ | $56.20 \pm 0.31$ |
| Random ($p = 0.9$) | $\|\mathcal{D}\|$ | $2\|\mathcal{D}\|$ | $59.71 \pm 0.52$ | $57.10 \pm 0.66$ | $55.54 \pm 0.62$ |
| Random ($p = 1.0$) | $\|\mathcal{D}\|$ | $2\|\mathcal{D}\|$ | $59.32 \pm 0.85$ | $57.49 \pm 0.24$ | $56.17 \pm 0.74$ |
| WoN ($N = 4$) | $\|\mathcal{D}\|/2$ | $2\|\mathcal{D}\|$ | $60.34 \pm 1.09$ | $58.19 \pm 1.07$ | $56.61 \pm 0.24$ |
| WoN ($N = 8$) | $\|\mathcal{D}\|/4$ | $2\|\mathcal{D}\|$ | $\underline{60.64} \pm 0.61$ | $58.03 \pm 0.56$ | $56.00 \pm 0.62$ |
| WoN ($N = 128$) | $\|\mathcal{D}\|/64$ | $2\|\mathcal{D}\|$ | $51.55 \pm 0.53$ | $52.88 \pm 0.20$ | $50.16 \pm 0.16$ |
| Coreset | $\|\mathcal{D}\|$ | $2\|\mathcal{D}\|$ | $56.71 \pm 0.93$ | $57.67 \pm 0.52$ | $56.57 \pm 0.20$ |
| Perplexity | $\|\mathcal{D}\|$ | $2\|\mathcal{D}\|$ | $60.05 \pm 0.52$ | $57.91 \pm 1.05$ | $54.23 \pm 0.56$ |
| AEPO ($\lambda = 0$) | $\|\mathcal{D}\|$ | $2\|\mathcal{D}\|$ | $56.83 \pm 0.49$ | $55.26 \pm 1.05$ | $54.92 \pm 0.16$ |
| AEPO ($\lambda = 0.5$) | $\|\mathcal{D}\|$ | $2\|\mathcal{D}\|$ | $59.23 \pm 0.91$ | $\mathbf{60.31} \pm 0.16$ | $56.42 \pm 0.31$ |
| AEPO ($\lambda = 1.0$) | $\|\mathcal{D}\|$ | $2\|\mathcal{D}\|$ | $\mathbf{62.40} \pm 0.22$ | $\underline{60.29} \pm 0.50$ | $\underline{56.97} \pm 0.24$ |
| AEPO ($\lambda = 2.0$) | $\|\mathcal{D}\|$ | $2\|\mathcal{D}\|$ | $59.71 \pm 0.45$ | $59.79 \pm 0.95$ | $\mathbf{57.36} \pm 0.38$ |
| WoN ($N = 128$) | $\|\mathcal{D}\|$ | $128\|\mathcal{D}\|$ | $70.12 \pm 0.56$ | $67.31 \pm 0.25$ | $64.17 \pm 0.66$ |

Table 14: Evaluation of AEPO on Anthropic's Helpfulness dataset using Mistral. The mean and standard deviation of three runs are shown. Note that OASST is used as a proxy reward model to annotate the preference of the training dataset.

| Preference Dataset Configuration | | | | |
|---|---|---|---|---|
| Method | #Insts | #Annots | OASST | Eurus |
| SFT (Mistral) | 0 | 0 | 4.690 | 1311.75 |
| Random ($p = 0.9$) | $|\mathcal{D}|$ | $2|\mathcal{D}|$ | $5.182 \pm 0.017$ | $1570.70 \pm 14.68$ |
| WoN ($N = 4$) | $|\mathcal{D}|/2$ | $2|\mathcal{D}|$ | $5.131 \pm 0.021$ | $1566.81 \pm 11.38$ |
| WoN ($N = 8$) | $|\mathcal{D}|/4$ | $2|\mathcal{D}|$ | $5.170 \pm 0.008$ | $\underline{1609.48} \pm 4.32$ |
| AEPO ($\lambda = 0.5$) | $|\mathcal{D}|$ | $2|\mathcal{D}|$ | $\mathbf{5.255} \pm 0.018$ | $\mathbf{1702.30} \pm 9.405$ |
| AEPO ($\lambda = 1.0$) | $|\mathcal{D}|$ | $2|\mathcal{D}|$ | $5.177 \pm 0.008$ | $1582.73 \pm 12.53$ |
| AEPO ($\lambda = 2.0$) | $|\mathcal{D}|$ | $2|\mathcal{D}|$ | $\underline{5.219} \pm 0.011$ | $1599.03 \pm 18.620$ |
| WoN ($N = 128$) | $|\mathcal{D}|$ | $128|\mathcal{D}|$ | $5.186 \pm 0.007$ | $1648.45 \pm 7.56$ |

Table 15: Win rate against the SFT model on Anthropic's Helpfulness dataset. The mean and standard deviation of three runs are shown. Note that OASST is used as a proxy reward model to annotate the preference of the training dataset.

| Preference Dataset Configuration | | | | | |
|---|---|---|---|---|---|
| Method | #Insts | #Annots | OASST (w%) | Eurus (w%) | PairRM (w%) |
| SFT (Mistral) | 0 | 0 | 50 | 50 | 50 |
| Random ($p = 0.9$) | $|\mathcal{D}|$ | $2|\mathcal{D}|$ | $66.02 \pm 0.65$ | $61.48 \pm 0.36$ | $\underline{60.67} \pm 0.81$ |
| WoN ($N = 4$) | $|\mathcal{D}|/2$ | $2|\mathcal{D}|$ | $64.31 \pm 0.84$ | $62.13 \pm 0.48$ | $59.71 \pm 0.27$ |
| WoN ($N = 8$) | $|\mathcal{D}|/4$ | $2|\mathcal{D}|$ | $66.39 \pm 0.14$ | $63.04 \pm 0.43$ | $60.53 \pm 0.30$ |
| AEPO ($\lambda = 0.5$) | $|\mathcal{D}|$ | $2|\mathcal{D}|$ | $\mathbf{68.02} \pm 1.04$ | $\mathbf{67.99} \pm 0.52$ | $\mathbf{61.78} \pm 0.26$ |
| AEPO ($\lambda = 1.0$) | $|\mathcal{D}|$ | $2|\mathcal{D}|$ | $\underline{66.81} \pm 0.36$ | $62.06 \pm 0.50$ | $59.50 \pm 0.31$ |
| AEPO ($\lambda = 2.0$) | $|\mathcal{D}|$ | $2|\mathcal{D}|$ | $65.67 \pm 0.26$ | $\underline{63.77} \pm 0.90$ | $59.49 \pm 0.29$ |
| WoN ($N = 128$) | $|\mathcal{D}|$ | $128|\mathcal{D}|$ | $66.06 \pm 0.29$ | $65.31 \pm 0.32$ | $61.40 \pm 0.15$ |

Table 16: Evaluation of AEPO on Anthropic's Harmlessness dataset using Mistral. The mean and standard deviation of three runs are shown. Note that OASST is used as a proxy reward model to annotate the preference of the training dataset.

| Preference Dataset Configuration | | | | |
|---|---|---|---|---|
| Method | #Insts | #Annots | OASST | Eurus |
| SFT (Mistral) | 0 | 0 | -1.291 | -43.87 |
| Random ($p = 0.9$) | $|\mathcal{D}|$ | $2|\mathcal{D}|$ | $-0.024 \pm 0.003$ | $433.93 \pm 5.00$ |
| WoN ($N = 4$) | $|\mathcal{D}|/2$ | $2|\mathcal{D}|$ | $0.001 \pm 0.021$ | $446.87 \pm 4.66$ |
| WoN ($N = 8$) | $|\mathcal{D}|/4$ | $2|\mathcal{D}|$ | $-0.376 \pm 0.019$ | $313.01 \pm 10.18$ |
| AEPO ($\lambda = 0.5$) | $|\mathcal{D}|$ | $2|\mathcal{D}|$ | $\underline{0.632} \pm 0.031$ | $\mathbf{779.87} \pm 7.61$ |
| AEPO ($\lambda = 1.0$) | $|\mathcal{D}|$ | $2|\mathcal{D}|$ | $0.121 \pm 0.002$ | $502.79 \pm 14.87$ |
| AEPO ($\lambda = 2.0$) | $|\mathcal{D}|$ | $2|\mathcal{D}|$ | $\mathbf{0.665} \pm 0.023$ | $\underline{685.82} \pm 15.55$ |
| WoN ($N = 128$) | $|\mathcal{D}|$ | $128|\mathcal{D}|$ | $0.071 \pm 0.010$ | $530.02 \pm 3.65$ |

Table 17: Win rate against the SFT model (Mistral) on Anthropic's Harmlessness dataset. The mean and standard deviation of three runs are shown. Note that OASST is used as a proxy reward model to annotate the preference of the training dataset.

| Preference Dataset Configuration | | | | | |
|---|---|---|---|---|---|
| Method | #Insts | #Annots | OASST (w%) | Eurus (w%) | PairRM (w%) |
| SFT (Mistral) | 0 | 0 | 50 | 50 | 50 |
| DPO ($p = 0.9$) | $|\mathcal{D}|$ | $2|\mathcal{D}|$ | $71.10 \pm 0.26$ | $68.30 \pm 0.09$ | $67.51 \pm 0.33$ |
| WoN ($N = 4$) | $|\mathcal{D}|/2$ | $2|\mathcal{D}|$ | $72.45 \pm 0.34$ | $69.43 \pm 0.15$ | $67.71 \pm 0.93$ |
| WoN ($N = 8$) | $|\mathcal{D}|/4$ | $2|\mathcal{D}|$ | $66.97 \pm 0.43$ | $64.21 \pm 0.51$ | $64.53 \pm 0.34$ |
| AEPO ($\lambda = 0.5$) | $|\mathcal{D}|$ | $2|\mathcal{D}|$ | $\underline{79.47} \pm 0.47$ | $\mathbf{80.13} \pm 0.46$ | $\mathbf{69.72} \pm 0.59$ |
| AEPO ($\lambda = 1.0$) | $|\mathcal{D}|$ | $2|\mathcal{D}|$ | $73.79 \pm 0.13$ | $71.62 \pm 0.71$ | $\underline{68.76} \pm 0.09$ |
| AEPO ($\lambda = 2.0$) | $|\mathcal{D}|$ | $2|\mathcal{D}|$ | $\mathbf{80.55} \pm 0.09$ | $\underline{77.65} \pm 0.62$ | $67.87 \pm 0.85$ |
| WoN ($N = 128$) | $|\mathcal{D}|$ | $128|\mathcal{D}|$ | $72.72 \pm 0.25$ | $72.54 \pm 0.17$ | $68.27 \pm 0.32$ |

Table 18: Evaluation of preference dataset configuration strategies for off-policy learning. We generate responses using Mistral and use the generated responses to train Dolly. LoRA hyperparameters are set $r = 32$ and $\alpha = r/4$. Note that OASST is used as a proxy reward model to annotate the preference of the training dataset.

| Preference Dataset Configuration | | | | | | | |
|---|---|---|---|---|---|---|---|
| Method | #Insts | #Annots | OASST | Eurus | OASST (w%) | Eurus (w%) | PairRM (w%) |
| SFT (Dolly) | 0 | 0 | -1.837 | -1275.06 | 50 | 50 | 50 |
| Random ($p = 0.8$) | $|\mathcal{D}|$ | $2|\mathcal{D}|$ | -1.672 | $\underline{-1206.83}$ | 55.53 | 52.11 | 53.19 |
| Random ($p = 0.9$) | $|\mathcal{D}|$ | $2|\mathcal{D}|$ | -1.682 | -1213.65 | 54.41 | 51.97 | $\mathbf{54.08}$ |
| Random ($p = 1.0$) | $|\mathcal{D}|$ | $2|\mathcal{D}|$ | -1.685 | -1232.98 | 52.42 | 51.08 | 52.19 |
| WoN ($N = 4$) | $|\mathcal{D}|/2$ | $2|\mathcal{D}|$ | -1.664 | -1221.01 | 53.17 | 51.71 | 53.80 |
| WoN ($N = 8$) | $|\mathcal{D}|/4$ | $2|\mathcal{D}|$ | -1.700 | -1233.16 | 52.92 | 50.99 | 53.00 |
| WoN ($N = 128$) | $|\mathcal{D}|/64$ | $2|\mathcal{D}|$ | -1.794 | -1255.30 | 50.87 | 49.72 | 49.35 |
| AEPO ($\lambda = 0$) | $|\mathcal{D}|$ | $2|\mathcal{D}|$ | -1.786 | -1248.58 | 51.12 | 50.03 | 50.54 |
| AEPO ($\lambda = 0.5$) | $|\mathcal{D}|$ | $2|\mathcal{D}|$ | -1.609 | -1208.81 | $\underline{55.78}$ | 52.34 | 53.75 |
| AEPO ($\lambda = 1.0$) | $|\mathcal{D}|$ | $2|\mathcal{D}|$ | $\mathbf{-1.555}$ | $\mathbf{-1177.69}$ | 55.40 | $\mathbf{53.95}$ | $\underline{53.92}$ |
| AEPO ($\lambda = 2.0$) | $|\mathcal{D}|$ | $2|\mathcal{D}|$ | $\underline{-1.590}$ | -1207.26 | $\mathbf{56.89}$ | $\underline{52.53}$ | 52.89 |
| WoN ($N = 128$) | $|\mathcal{D}|$ | $128|\mathcal{D}|$ | -1.409 | -1140.61 | 60.50 | 56.02 | 56.44 |

## H.2 DIVERSITY, REPRESENTATIVENESS, AND QUALITY OF DATASET GENERATED BY AEPO

Figures 12, 13, and 14 show the diversity (pairwise sentence BERT and distinct-n) and representativeness of the preference dataset $\mathcal{D}_{AE}$ generated by AEPO on AlpacaFarm and hh-rlhf datasets. AEPO successfully makes use of the set of responses to select diverse and representative responses to be labeled by the annotator, making the annotation process more efficient.

Figures 15, 16, and 17 show the diversity (distinct-n) and quality (mean reward) tradeoff. AEPO successfully improves the diverse-quality tradeoff with a larger number of response texts.

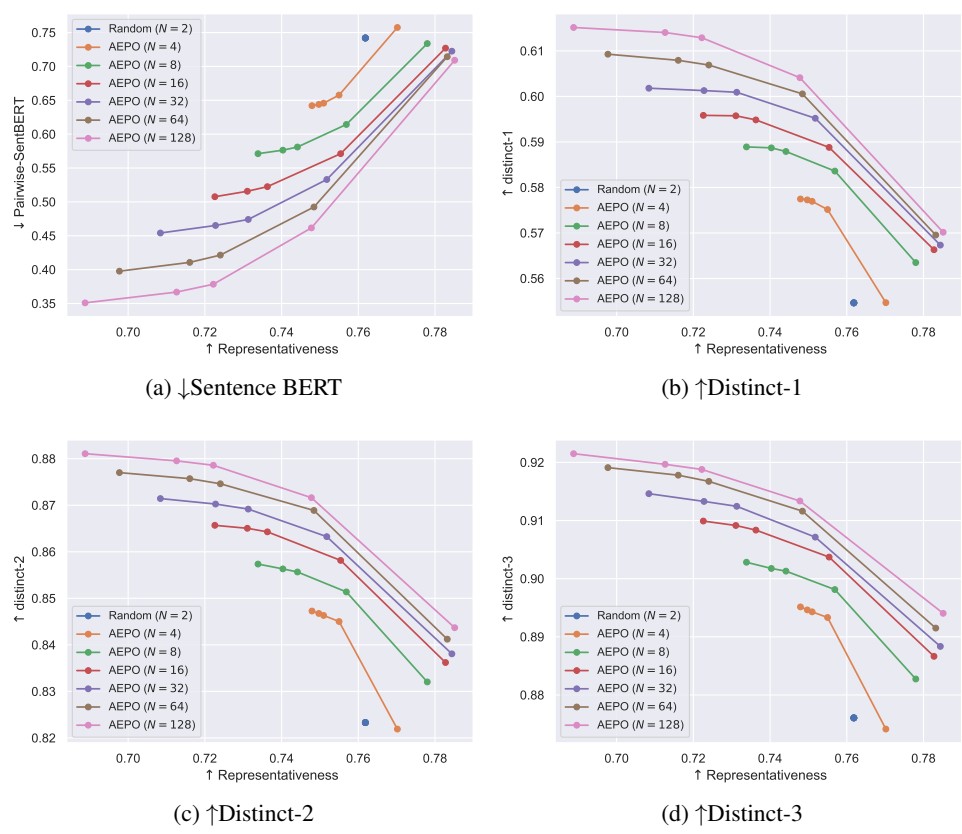

Figure 12: Diversity (↓Sentence BERT and ↑Distinct-n) and representativeness of the responses of the preference datasets $\mathcal{D}_{AE}$ generated by AEPO with different numbers of input responses. AEPO successfully generates datasets with better diversity-representativeness tradeoffs.

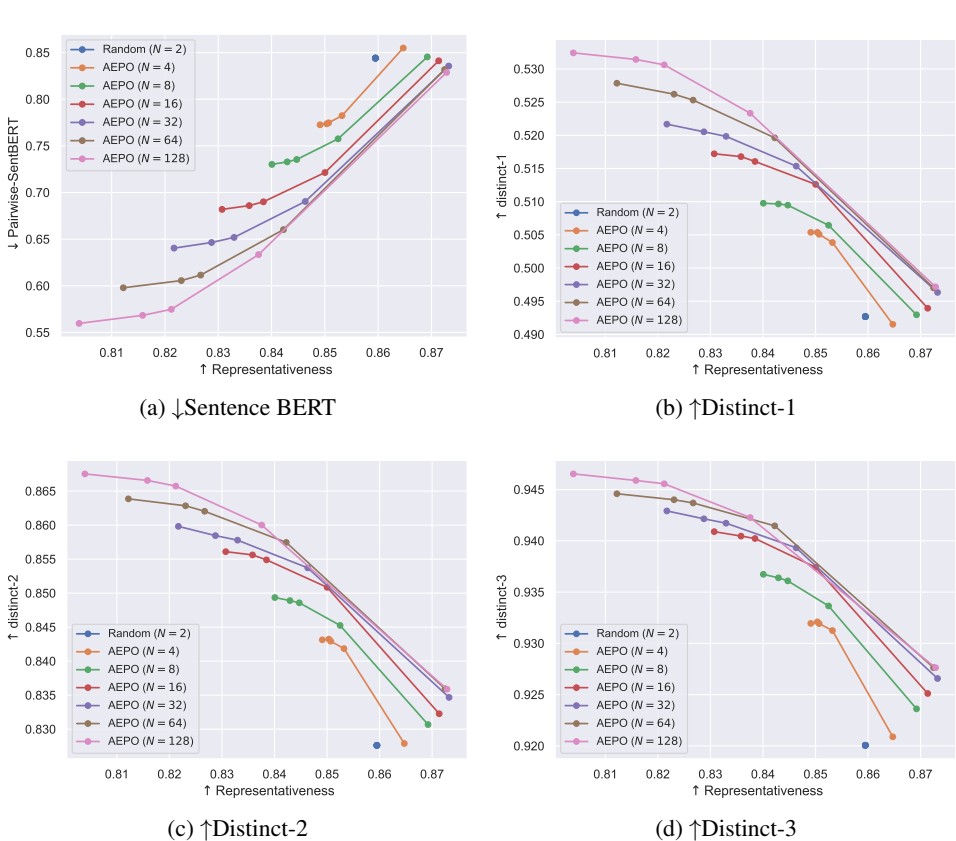

Figure 13: Diversity (↓Sentence BERT and ↑Distinct-n) and representativeness of the responses of the preference datasets $\mathcal{D}_{AE}$ generated by AEPO with different numbers of input responses on Anthropic's Helpfulness dataset.

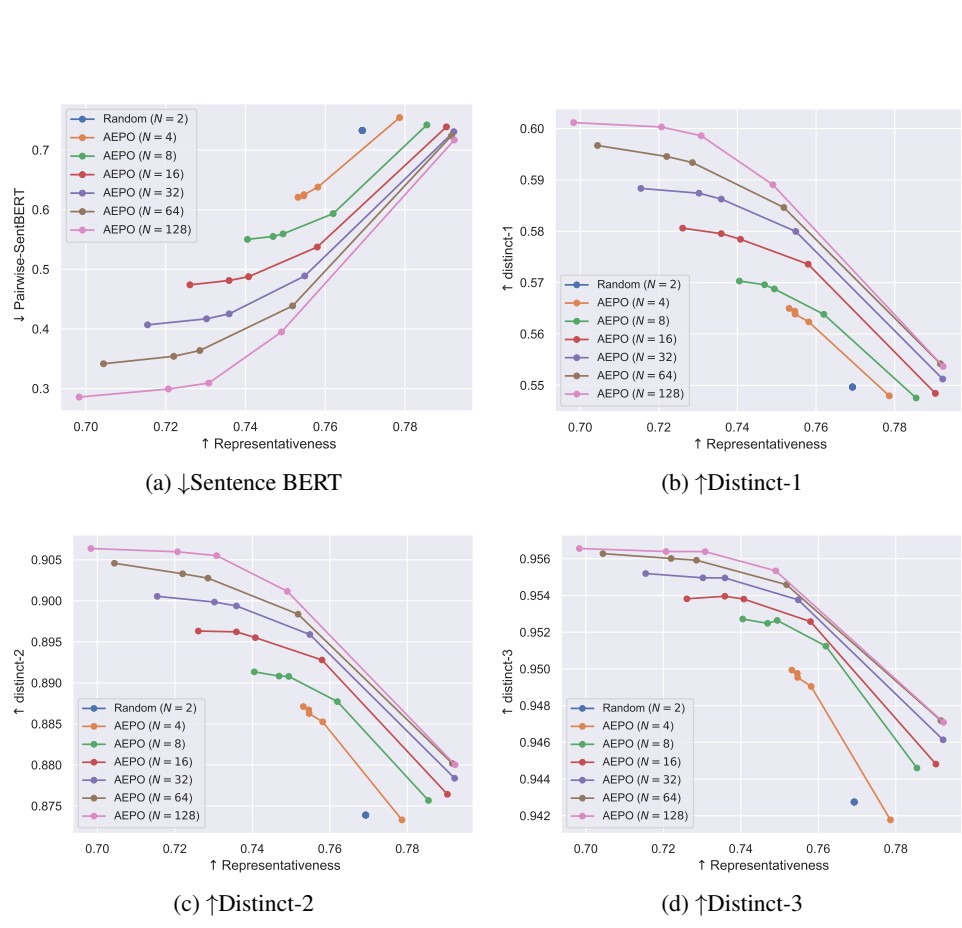

(a) ↓Sentence BERT

(b) ↑Distinct-1

(c) ↑Distinct-2

(d) ↑Distinct-3

Figure 14: Diversity (↓Sentence BERT and ↑Distinct-n) and representativeness of the responses of the preference datasets $\mathcal{D}_{AE}$ generated by AEPO with different numbers of input responses on Anthropic's Harmlessness dataset.

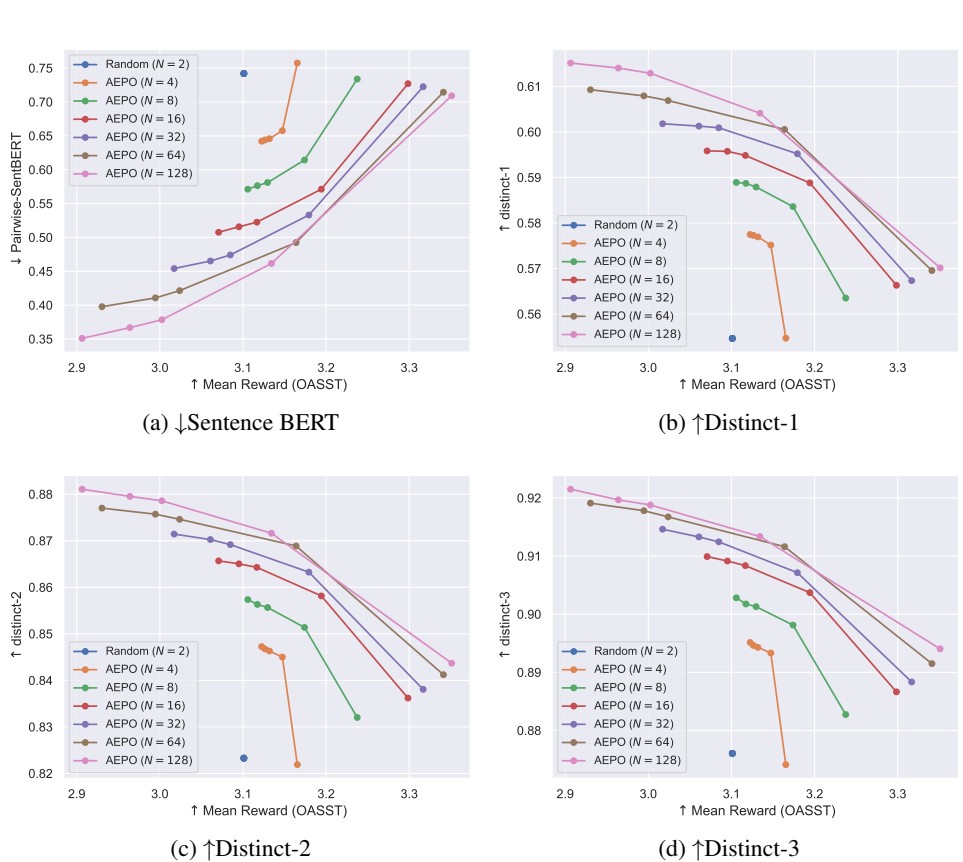

Figure 15: Diversity (↓Sentence BERT and ↑Distinct-n) and quality (↑mean reward) of the responses of the preference datasets $\mathcal{D}_{AE}$ generated by AEPO with different numbers of input responses. AEPO successfully generates datasets with better diversity-quality tradeoffs.

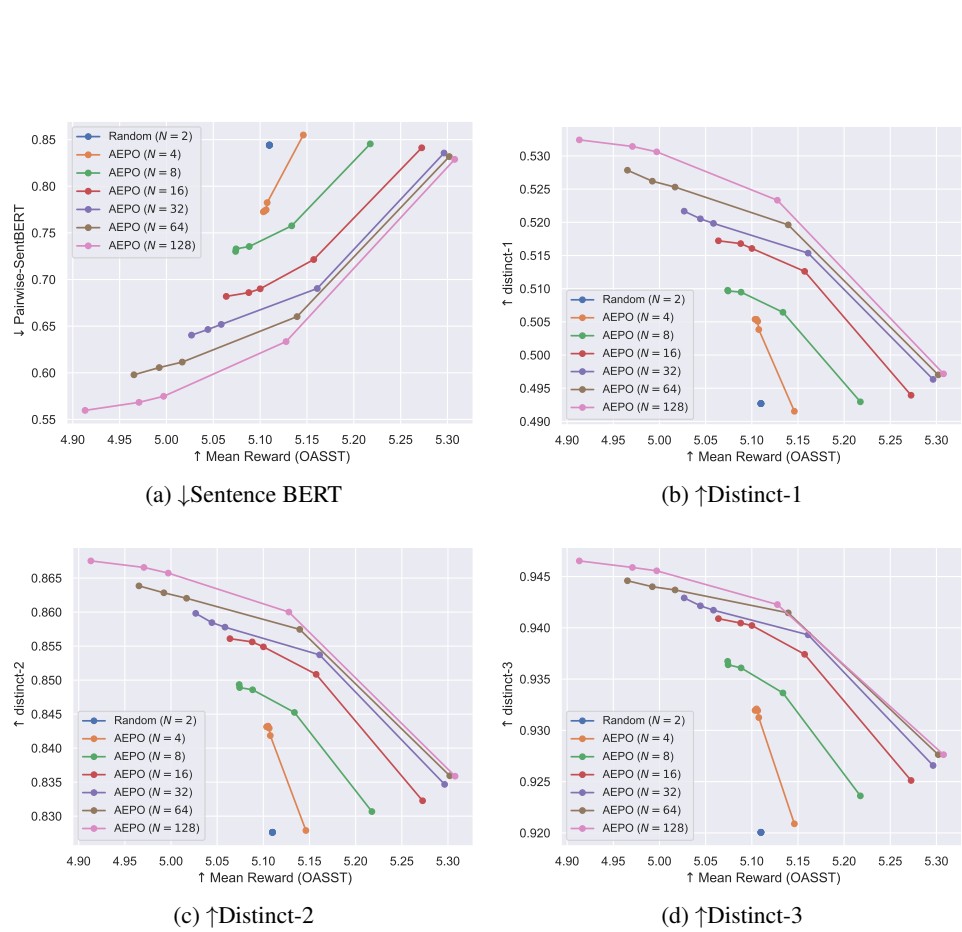

(a) ↓Sentence BERT

(b) ↑Distinct-1

(c) ↑Distinct-2

(d) ↑Distinct-3

Figure 16: Diversity (↓Sentence BERT and ↑Distinct-n) and quality (↑mean reward) of the responses of the preference datasets $\mathcal{D}_{AE}$ generated by AEPO with different numbers of input responses on Anthropic's Helpfulness dataset.

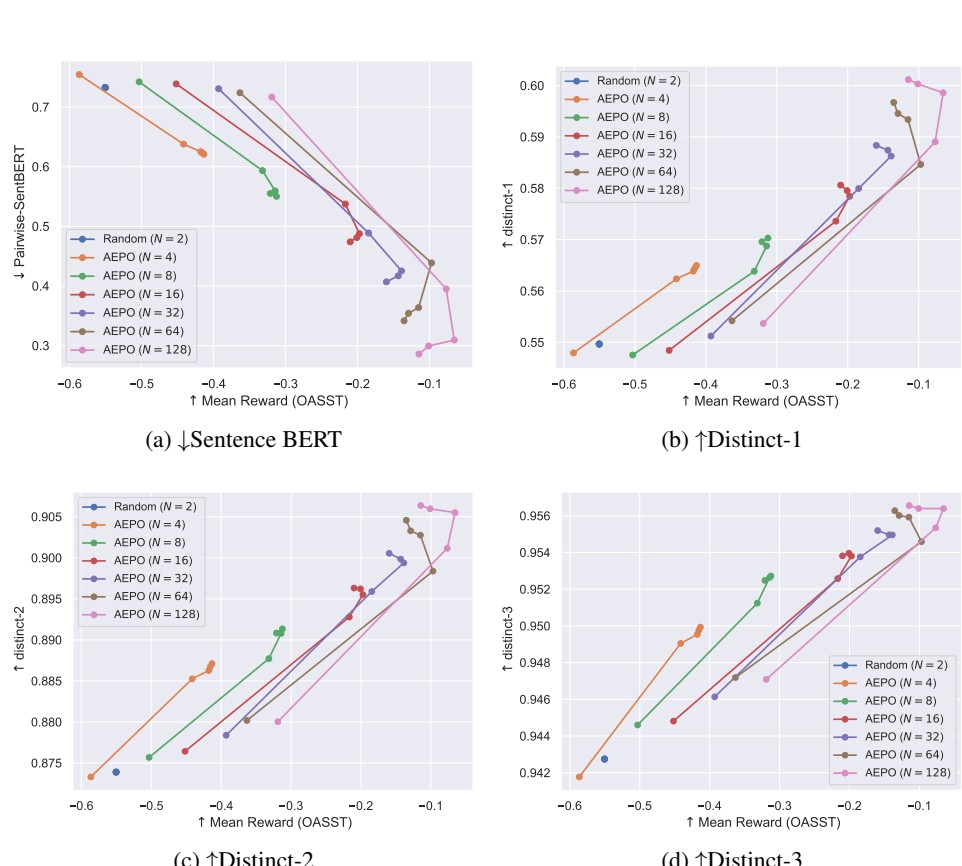

Figure 17: Diversity (↓Sentence BERT and ↑Distinct-n) and quality (↑mean reward) of the responses of the preference datasets $\mathcal{D}_{AE}$ generated by AEPO with different numbers of input responses on Anthropic's Harmlessness dataset.

## I  LIMITATIONS

Although our method is motivated by the situation where the annotation is needed to align the language model, the majority of our experiments (AlpacaFarm and Anthropic's hh-rlhf) are conducted using a proxy reward model to annotate preference on training datasets instead of using human annotation. We use human annotation for the JCM dataset but use an LLM to automatically evaluate the agreement of the response text with the human annotation. Manual human annotation would be desirable for future work.

Our focus is on developing a method to generate a diverse and representative set of responses. The preparation of diverse and representative instructions is also an important task to generate an efficient dataset (Sanh et al., 2022; Ding et al., 2023; Cui et al., 2023; Liu et al., 2024a; Xu et al., 2024a). Our method is orthogonal to methods for generating high quality instructions and can be combined. Comparing and combining AEPO with methods for generating diverse instructions is future work.

All experiments are performed using LoRA (Hu et al., 2022). The evaluation of AEPO with full parameter fine-tuning is future work. Our experiments are limited to the evaluation on DPO. Evaluating AEPO on variants of DPO (Amini et al., 2024; Gheshlaghi Azar et al., 2024; Tang et al., 2024b; Morimura et al., 2024; Zhang et al., 2024b) and other preference optimization algorithms (Ouyang et al., 2022; Zhao et al., 2023; Ahmadian et al., 2024) is future work.

The performance of AEPO depends on the choice of the hyperparameter $\lambda$. We observe that $\lambda = 1.0$ is a good choice throughout the experiments, but developing a strategy to find an effective $\lambda$ for a given dataset is future work.

## J  COMPUTATIONAL RESOURCES

Text generation and DPO training run on an instance with an NVIDIA A100 GPU with 80 GB VRAM, 16 CPU cores, and 48 GB memory. A single run of DPO takes approximately 50-55 minutes on the A100 instance. AEPO runs on an NVIDIA A2 GPU with 8 GB VRAM, 8 CPU cores, and 24 GB memory. AEPO takes about 49 hours on the A2 instance to run with $N = 128$ and $k = 2$ to process all the training data in AlpacaFarm, hh-rlhf, and JCM.

All the experiments are run using Huggingface's Transformers library (Wolf et al., 2020) and Transformer Reinforcement Learning library (von Werra et al., 2020).

## K  REPRODUCIBILITY STATEMENT

All the datasets and models used in the experiments are publically accessible (Table 19) except for GPT-4. Our code will be available on acceptance as an open source.

## L  IMPACT STATEMENT

We believe that this work will have a positive impact by encouraging work on AI systems that work better with a diverse set of people. LLMs would be more useful if they could adapt to the preferences of diverse groups of people, even if little preference annotation is available from their communities.

We foresee our method being useful for personalizing LLMs (Greene et al., 2023; Jang et al., 2023; Kirk et al., 2023). Personalized LLMs could have far-reaching benefits, but also a number of worrisome risks, such as the propagation of polarized views. We refer to Kirk et al. (2023) for a discussion of potential risks and countermeasures for personalized LLMs.

Table 19: List of datasets and models used in the experiments.

| Name | Reference |
| --- | --- |
| AlpacaFarm | Dubois et al. (2023) `https://huggingface.co/datasets/tatsu-lab/alpaca_farm` |
| Anthropic's hh-rlhf | Bai et al. (2022) `https://huggingface.co/datasets/Anthropic/hh-rlhf` |
| JCommonsenseMorality | Takeshita et al. (2023) `https://github.com/Language-Media-Lab/commonsense-moral-ja` |
| mistral-7b-sft-beta (Mistral) | Jiang et al. (2023a); Tunstall et al. (2024) `https://huggingface.co/HuggingFaceH4/mistral-7b-sft-beta` |
| dolly-v2-3b (Dolly) | Conover et al. (2023) `https://huggingface.co/databricks/dolly-v2-3b` |
| calm2-7b-chat (CALM2) | `https://huggingface.co/cyberagent/calm2-7b-chat` |
| OASST | Köpf et al. (2023) `https://huggingface.co/OpenAssistant/reward-model-deberta-v3-large-v2` |
| PairRM | Jiang et al. (2023b) `https://huggingface.co/llm-blender/PairRM` |
| Eurus | Yuan et al. (2024a) `https://huggingface.co/openbmb/Eurus-RM-7b` |
| Gemma2-9B | Team et al. (2024) `https://huggingface.co/google/gemma-2-9b-it` |
| Gemma2-27B | Team et al. (2024) `https://huggingface.co/google/gemma-2-27b-it` |
| MPNet | Song et al. (2020) `https://huggingface.co/sentence-transformers/all-mpnet-base-v2` |

