# OpenReview forum: "Annotation-Efficient Language Model Alignment via Diverse and Representative Response Texts"
_ICLR.cc/2025/Conference — ICLR 2025 Conference Withdrawn Submission_

### Official Review · Reviewer_qpRX · 2024-10-29

**Soundness:** 2
**Presentation:** 3
**Contribution:** 2
**Rating:** 3
**Confidence:** 4

**Summary:**

This paper proposes a strategy, annotation-efficient preference optimization (AEPO), to select a subset of preference pairs to be sent for annotation. Collecting human annotations is costly, therefore, if we can select only a subset of preference pairs and achieve close performance to that of annotating the whole set it can cut down costs significantly. The authors propose to select a set of responses from the pool of candidate pairs that have the highest information gain. They propose two heuristics for selecting the subset that might lead to a higher information gain: (1) selected elements should be, on average, close to all candidate pairs, and (2) selected elements should exhibit diversity.  AEPO is benchmarked against other subsampling strategies by using the subsampled data with DPO and evaluate the performance of the aligned model.

**Strengths:**

The motivation behind this paper is highly relevant to the research community. In scenarios where human annotations are costly, reducing the number of annotations can substantially cut down expenses. Leveraging information gain as a criterion for sample selection in the context of alignment presents a promising approach.

**Weaknesses:**

**Methodology:** There is a type mismatch between the definition of information gain (which applies to sets of responses) and its application to preference pairs (which are sets of response pairs). Additionally, the paper introduces information gain broadly but quickly shifts to the two heuristics without a formal mathematical connection between them. A constructive addition would be to use toy examples to measure information gain and demonstrate how the two heuristic metrics correlate with it.

Maybe I have missed it, but the authors mention they optimize Eq. 18, without mentioning how they do that. Does this involve iterating all N choose k pairs and picking the best one? How efficient is the selection algorithm? Discussing the efficiency of the selection algorithm is an important factor that is missing in the paper.

**Implementation:** The paper distances itself from approaches focused on generating quality synthetic data, aiming instead to design a subsampling strategy that minimizes human annotation requirements. Given this, the use of a reward model in place of human preferences in the experiments does not make sense. If the primary objective is to evaluate the algorithm's efficiency in selecting representative samples for human annotation, direct use of human preferences appears essential. Alternatively, one could consider using a large reward model as a proxy for human judgment and a smaller reward model for methods like West-of-N that are specifically for synthetic data generation.

Similarly, the comparison to West-of-N may not be entirely fair. The authors use fewer prompts for West-of-N, which could unfairly disadvantage its performance; it’s possible that West-of-N is highly effective, and the reduced prompts are the primary factor impacting results. Figure 11 in the appendix seems to support this interpretation. A potential solution is to use a reward model for West-of-N while using human annotation for AEPO, or to use a weaker reward model for West-of-N and a stronger one for annotating the final two AEPO candidates while using the same number of prompts for both methods.

**Questions:**

- How do you optimize Eq. 18?
- In line 188, it is mentioned that after AEPO, West-of-N is applied to select 2 candidates, is this necessary? Is there ablations on this?

---

### Official Review · Reviewer_Hp4d · 2024-11-03

**Soundness:** 4
**Presentation:** 4
**Contribution:** 3
**Rating:** 8
**Confidence:** 4

**Summary:**

This paper develops a method called Annotation-Efficient Preference Optimization (AEPO) for creating more efficient and effective preference datasets by subsampling responses using criteria that seek to maximize diversity and representativeness, where these quantities are estimated based on embedding similarity. The experiments use Mistral-7b-sft-beta as the model, DPO as the core preference learning objective, and AlpacaFram and Anthropic's Helpfulness/Harmlessness as the datasets. The annotations are done by the OASST reward model. In these experiments, AEPO outperforms random sampling, and random sampling with only the best and worst kep (West of N). Ablation studies show that both the diversity and representativeness make important contributions to these results.

**Strengths:**

AEPO seems like a valuable technique targeting an important area, and the results show that being smart about which preference pairs to use can be more effective than simply having a lot of data, at least for the configurations studied in the paper.

The experiments seem to be carefully done and comprehensively reported, and the gains seem significant.

**Weaknesses:**

1. The opening of the paper emphasizes how important it is to use human annotators, so it is surprising to find, in section 4, that no human annotation was used for the experiments.

2. The above is part of a larger tweak to the narrative that I would suggest: there is evidently no reason to focus on cost in particular, since the expensive settings seem to do very poorly.

3. The choice of DPO as the objective is significant, I believe. In Figure 3, the blue line goes up a bit and then goes down as the number of responses increases. Evidently, this process is actually hurting. A good explanation of this is that the AlpacaFarm data are just overall much worse than the model (Mistral), so that even the winning AlpacaFarm examples are actually bad from the perspective of the model. This happens with DPO because DPO just wants the winner to be more likely than the loser -- it doesn't care what the likelihoods of either are before learning begins. If this is a correct explanation, then what AEPO is doing is not about cost but rather about quality. For different objectives than DPO, we might see a different pattern.

4. I can't find a version of Figure 3 for the Anthropic dataset. That would be significant because I suppose Mistral has the opposite relation to that dataset -- it is weaker than whatever model generated the Anthropic examples. Please do let me know if I have missed something.

5. The Related work is pretty sparse. I suppose we can give a pass, but it would be good to at least cite other work that has sought to improve these preference datasets. [Wang et al. 2024](https://arxiv.org/abs/2407.16216) is a valuable survey of methods in this area -- it could be cited, as could closely related papers it cites.

**Questions:**

1. The sum in (13) is a bit surprising. Isn't this misaligned with the intuitive goal of distinctness. A group of very similar examples with one extraordinarily dissimilar one could get a higher distance sum here than a group of uniformly quite dissimilar examples. Does this matter? I am not sure whether this can arise, and it could be that the min/max sample step in (2) makes it irrelevant.

2. The paper says on line 144 that the dataset is the main factor in outcomes in this space. What is the evidence for this, as opposed to tracing results to interactions between the dataset, the model, and the objective?

---

> ### Comment · Reviewer_Hp4d · 2024-11-26
> **A nice paper; no author responses**
>
> This paper offers what seems to be a valuable technique. I would have been interested to see the authors' responses, especially on the theme of how this proposal relates to prior work.

---

### Official Review · Reviewer_KU9c · 2024-11-04

**Soundness:** 2
**Presentation:** 3
**Contribution:** 2
**Rating:** 5
**Confidence:** 4

**Summary:**

Authors extended DMBR decoding into preference annotations and propose annotation-efficient preference optimization (AEPO). AEPO annotates a subset of responses that maximize diversity and representatives from the available responses. AEPO was emperically shown to outperform DPO under the same annotation budget.

**Strengths:**

1. Annotation efficiency is an important research problem in preference learning.

2. AEPO empirically has some improvements over DPO or DPO variant.

**Weaknesses:**

1. The AEPO approach appears to be a straightforward extension of Diverse Minimum Bayes Risk (DMBR) decoding (https://arxiv.org/pdf/2401.05054) applied to preference annotations. The objective function remains largely identical to that in DMBR decoding.

2. Lack of Comparative Baseline with DMBR: Since the DMBR paper already claims to improve data quality in generated responses, it’s essential for the authors to evaluate AEPO directly against DMBR decoding. This would clarify whether AEPO actually enhances annotation efficiency or simply benefits from improvements in data quality.

3. Hyperparameter Sensitivity and Generalization: The model’s performance appears to be highly sensitive to the choice of λ, as shown in Figure 5. This heavy dependence on tuning raises concerns about the method’s generalizability across different tasks and datasets, as selecting an optimal λ may require extensive trial and error.

4. Lack of baselines using two selection principles: If you want to focus your contributions on your two data selection principles. You should at least explore more implementations. For example, you can do diversity beam search and apply MBR as a baseline. To prove those two principles are general and not constraint on a single implementation.

**Questions:**

See weakness

Consider to cite those relevant works:
https://arxiv.org/abs/2402.00396
https://arxiv.org/abs/2406.12168

---

### Official Review · Reviewer_iQDo · 2024-11-04

**Soundness:** 1
**Presentation:** 2
**Contribution:** 1
**Rating:** 3
**Confidence:** 5

**Summary:**

The paper proposes to select generations for preference annotation by optimizing an objective that combines representativeness (in the sense of the selected generations being similar on average to the full set of generations) and diversity (in the sense of the subset of selected generations being dissimilar to each other). This method is compared primarily with "West-of-N", in which preferences are annotated between all pairs of generations.

**Strengths:**

The authors identify an important problem, which is the need to make best possible of use of human preference annotation effort. The paper offers a broad range of evaluations and a comprehensive discussion of the recent preference optimization literature. The method is described clearly and would be straightforward to implement, although the authors also make code available.

For me, the most persuasive result is AEPO at N>2 vs AEPO at N=2, which shows that the proposed method of example selection offers some improvement over naively sampling two random responses.

**Weaknesses:**

From a conceptual perspective, the idea of selecting diverse *and* representative training examples is not original, and the application to preference learning does not change the fundamentals in a significant way. See, e.g. (Wei et al 2006 "Submodularity in Data Subset Selection and Active Learning"; Bıyık et a; 2019 "Batch active learning using determinantal point processes"), which propose more principled approaches to the same goal. In fact, the situation is significantly simplified in the submission because the subset size is limited to 2, so that it is possible to completely enumerate the maximization over subsets in equation 18. A related point is that when selecting larger subsets, the approach of *averaging* over all similarities/distances may not be meaingful. Instead, it may be better to ensure that every point in Y^cand is well-covered by at least one example, but there is no need to cover it more than once. This is the intuition behind the Coreset baseline (which I was glad to see included), but it only becomes meaningful for larger subsets.

Empirically, the comparison with West-of-N balances the number of preference annotations (which is reasonable) by offering WoN dramatically fewer instructions. It is unsurprising that WoN does worse when offered up to 64 times fewer instruction examples, and it seems likely that for any fixed annotation budget B, we would do best by offering B/2 instructions with exactly two generations each.

Theoretically, the approach is motivated by an information theoretic argument in section 3. However, without a probability model underlying H[R], this argument is too informal to shed much light on the proposed approach.

**Questions:**

Would it be possible to apply this approach to selecting *instructions* as well as generations? That might yield a more interesting and useful algorithm.

What is the size of the candidate set for the results in figure 5?

---

### Note · Authors · 2024-12-16

**Comment:**

We appreciate the reviewers' feedback and have decided to withdraw the manuscript. After discussing among ourselves, we agree that the paper needs further improvement.
Our primary goal was to develop a practical method, with the propositions being minor additional enhancements. We will revise the paper to clarify that the research focuses on the practical contribution.

**Withdrawal Confirmation:**

I have read and agree with the venue's withdrawal policy on behalf of myself and my co-authors.